# ANU-RL: A New Perspective on Weakly-Supervised Representation Learning for Visual Place Recognition

## Abstract

Representation Learning (RL) is fundamental for image matching, retrieval, classification, and other applications, enabling task-specific feature learning. RL algorithms aim to learn compact embeddings that preserve the neighbourhood structure of the input data. A general approach to this is contrastive learning, which pulls similar images (positives) closer together and pushes dissimilar images (negatives) farther apart in the embedding space. In Visual Place Recognition (VPR), positive images of a query share specific geographical and visual attributes with the query and can, form a cluster. In contrast, negative images differ from the query and may vary among themselves or be similar. Most existing training objectives focus only on the relationships between query-positives and query-negatives. In this work, we hypothesize that, in addition to these relationships, other naturally available relationships, such as positives-to-negatives and intra-positives, can improve VPR performance by enhancing representation quality. The proposed framework, A New Perspective on Weakly-Supervised Representation Learning (ANU-RL), when integrated with VPR aggregators like BoQ, SALAD, MixVPR, and NetVLAD, achieves state-of-the-art performance on most challenging VPR benchmarks, including Pittsburgh 30k, Tokyo 24/7, Nordland, MSLS (val), and many others. Moreover, all of this comes at no extra cost at inference time. Further, we generalize the proposed framework to a wider range of metric learning applications, specifically image retrieval.

## 1 Introduction

Visual Place Recognition (VPR), also known as visual geolocalization, aims to predict the geographic location depicted in an input image. This is typically achieved by comparing a query image, whose location is unknown, with a database of geo-tagged reference images Lowry et al. (2015). Through this comparison, the best-matching reference image is identified for the given query. The geographic coordinates of this matched reference image are then assigned to the query image.

VPR is typically formulated as an image retrieval or classification problem that fundamentally relies on image matching. Image matching Ma et al. (2021) involves identifying and quantifying the similarity between a pair of images. This process is usually carried out in an embedding space, where task-specific feature learning plays a crucial role. In general, this is achieved by metric learning algorithms Kaya & Bilge (2019) that map the neighbourhood structure from the input space to the low-dimensional embedding space. This enables models to learn shared embeddings for similar images (similar geographically or perceptually) and distinct embeddings for dissimilar ones. The contrastive learning framework is a subset of such metric learning algorithms, aiming to minimize intra-class distance and maximize inter-class distance. In VPR, intra-class refers to a query and its positive neighbourhood from the same geographical location. In contrast, inter-class refers to negative samples around a query, where both query and its negatives are from distinct locations.

However, existing loss functions such as Multi-Similarity (Multi-Sim or MSim) loss Wang et al. (2019) consider relationships between query and positives, and query and negatives, ignoring other possible relationships between positives and negatives, and within positives. In this work, we propose a straightforward yet logical framework called A New perspective on Weakly-Supervised Representation Learning (ANU-RL) that

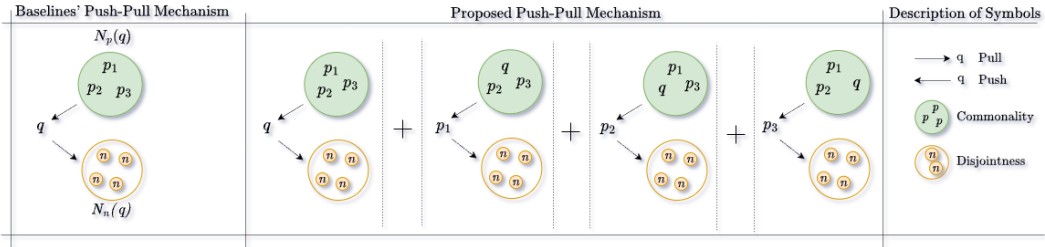

Figure 1: **The proposed push-pull mechanism**: The leftmost plot depicts the approach of the baseline, where this does not explicitly consider relationships among positives $\{p_1, p_2, p_3\}$. The figure in the middle illustrates the proposed approach (Eq. 2), which introduces an extra loop over all positives of a query ($q$) that treats each positive ($p_i$) as a query in each looping instance. The gradients computed over the aggregated ($+$) distance are used to update the model parameters in a training instance.

explicitly includes the proposed additional relationships in the existing loss functions, as illustrated in Figure 1. The proposed approach augments each query with its surrounding positives, forming a new set of queries $P'_q = P_q \cup \{q\}$. Each of the newly formed set of samples is pulled toward its peer-positive group and pushed away from less-relevant negative examples in the feature space. In this work, we apply the proposed framework to MSim loss Wang et al. (2019), Triplet loss Schroff et al. (2015), which are used by state-of-the-art VPR models Ali-bey et al. (2022; 2023); Arandjelovic et al. (2016); Wang et al. (2022); Zhu et al. (2023); Izquierdo & Civera (2024). We also extend this to the FastAP loss Cakir et al. (2019). The proposed approach can be applied to any triplet-based loss function. Figure 1 illustrates the ANU-RL framework for MSim loss.

The contributions of this work include:

1. The framework, ANU-RL, that explicitly considers the additional intra-positive and positive-negative relationships in the contrastive learning objectives.

2. Variants of ANU-RL framework: ANU-Easy (ANU-E) and ANU-Hard (ANU-H). While ANU-RL is ANU-ALL by default, where it incorporates all additional relationships, ANU-E (ANU-H) includes only the easiest (hardest) pair ignoring the other relationships. In other words, from Figure 1, instead of connecting $\{p_1, p_2, p_3\}$ to all examples, we retain only the easiest and hardest connection in green and orange circles. This is based on similarity scores.

3. An investigation into contrastive loss functions—such as MSim loss Wang et al. (2019), Triplet loss Schroff et al. (2015), and FastAP loss Cakir et al. (2019)—within the context of the proposed framework.

4. Implementation and comprehensive evaluation using multiple VPR aggregators, highlighting the enhanced representation quality achieved by the proposed method.

5. An analogy to the regularization technique.

6. Evaluating the generalization of the proposed framework for other relevant applications, such as Image retrieval (IR).

## 2 Related Work

### 2.1 Deep Metric Learning Algorithms

Most deep metric learning frameworks leverage the prior knowledge of training data, such as class labels, to define neighbourhood relationships between images that supervise the model training. Following this, various interesting training loss functions have been developed in the literature Hadsell et al. (2006); Hoffer

& Ailon (2015); Schroff et al. (2015); Ustinova & Lempitsky (2016); Rippel et al. (2015); Wang et al. (2017); Ge (2018). The Contrastive Loss Hadsell et al. (2006) optimizes pair-wise distances in the feature space. Triplet loss functions Hoffer & Ailon (2015); Schroff et al. (2015) argue that triplets perform better over pair-based learning. However, not all triplets may be informative and slow down the training convergence. To address this, the Triplet loss Schroff et al. (2015) introduces a mining strategy to sample informative pairs. These functions use only a single positive and a negative for each query during the optimization. In contrast, the N-Pair Loss Sohn (2016) introduces multiple negatives for each query for faster and better convergence. Further, Histogram Loss Ustinova & Lempitsky (2016) involves the distribution of similarities against simple Euclidean distance-based objectives. Magnet Loss Rippel et al. (2015) proposes a cluster-based optimization for better local neighbourhood structure. In-triplet mining approach Balntas et al. (2016) and Lifted-Structure loss Oh Song et al. (2016) consider an additional distance between positive and negative pairs to choose the hardest one. Further, Hermans et al. Hermans et al. (2017) generalizes Oh Song et al. (2016) to multiple positives and negatives of a query. Angular Loss Wang et al. (2017) introduces angle information between triplets. To reduce the training complexity, Proxy Anchor Loss Kim et al. (2020) learns proxies for each class and categorizes them as positives and negatives. Further, the Circle Loss Sun et al. (2020) proposes to weigh different pairs differently for better optimization. The SupCon Loss Khosla et al. (2020) extends the self-supervised Contrastive loss Chen et al. (2020) to a supervised version. Barbano et al. Barbano et al. (2022) introduces a framework to address the shortcomings Chen et al. (2020); Khosla et al. (2020). The FastAP loss Cakir et al. (2019) proposes distance quantization-based optimization. We observe that these objectives often implicitly assume that representations of similar images converge in the latent space, but none of them explicitly include an expression to achieve this. To fill this gap, we introduce a simple yet logical change that results in improved quality of representations, leading to improvement in several successful VPR models that are briefly discussed in section 2.2.

## 2.2 VPR Works

Among the loss functions discussed in Section 2.1, Triplet loss Schroff et al. (2015) and MSim loss Wang et al. (2019) are widely adopted in VPR Arandjelovic et al. (2016); Wang et al. (2022); Zhu et al. (2023); Ali-bey et al. (2022; 2023); Uggi & Channappayya (2024). Motivated by this, we evaluate our method primarily with Schroff et al. (2015); Wang et al. (2019), and additionally perform an ablation using the more recent FastAP loss Cakir et al. (2019). Most VPR approaches Jégou et al. (2010); Arandjelovic et al. (2016); Hausler et al. (2021); Ali-bey et al. (2023); Wang et al. (2022); Xu et al. (2023); Zhu et al. (2023); Lu et al. (2024a); Uggi & Channappayya (2025) are retrieval-based, where they focus on learning image-level descriptors, that are subsequently used for image retrieval. The VPR problem has a long history, starting from hand-designed approaches such as VLAD Jégou et al. (2010), which laid the groundwork for many subsequent deep learning-based methods. A typical VPR model generally consists of a backbone followed by an aggregator module. The VLAD layer Jégou et al. (2010) serves as such an aggregator, producing compact representations from the backbone features.

NetVLAD Arandjelovic et al. (2016) extends VLAD Jégou et al. (2010) by learning cluster centroids directly from data. Similarly, GeM Radenović et al. (2018) unifies the max and average pooling techniques. Building on NetVLAD Arandjelovic et al. (2016), Patch-NetVLAD Hausler et al. (2021) proposes a two-stage ranking strategy. MS-NetVLAD Uggi & Channappayya (2024) further exploits multi-scale features from different backbone layers and achieves notable performance gains through this simple modification. The Conv-AP Ali-bey et al. (2022) aggregates 3D feature block by channel-wise and spatial-wise dimensionality reduction. Unlike these techniques, the CosPlace and EigenPlaces Berton et al. (2022; 2023) pose VPR as a classification problem. MixVPR Ali-bey et al. (2023) proposes an all-MLP aggregator for feature mixing. TransVPR Wang et al. (2022) proposes a hybrid model composed of CNNs and ViT. R2Former Zhu et al. (2023) addresses the limitations of the classical RANSAC and improves it using attention scores. The state-of-the-art techniques, CricaVPR and SelaVPR Lu et al. (2024a;b), propose adapters to adapt the pretrained model to the downstream VPR task. SALAD Izquierdo & Civera (2024) reformulates NetVLAD Arandjelovic et al. (2016) using optimal transport between features to cluster distributions. Another recent work, BoQ Ali-Bey et al. (2024), introduces learnable global queries called bag of queries to capture the place specific features. While these approaches work with RGB images, the SNSM work Uggi & Channappayya (2025), a training-free aggregator, extracts domain invariant representation maps for RGB-IR cross-domain VPR.

## 3 Proposed ANU-RL Framework

Various existing contrastive learning objectives presented in section 2.1 are developed in the framework in Eq. 1,

$$\mathcal{L}_{original} = \frac{1}{|Q|} \sum_{q \in Q} \left( \sum_{k \in P_q} \mathcal{L}_{qk} + \sum_{l \in N_q} \mathcal{L}_{ql} \right), \tag{1}$$

where we see that the relationships utilized are limited to $q \in Q$ and its positives $P_q$, and $q$ and its negatives $N_q$. $Q$ is the set of queries and $|Q|$ is the cardinality, that is, the batch size in a training instance, $P_q$ ($N_q$) contains all positive (negative) samples corresponding to the query $q$.

In our framework,

$$\mathcal{L}_{anu-all} = \frac{1}{|Q|} \sum_{q \in Q} \sum_{p \in P'_q} \left( \sum_{k \in P'_q \backslash p} \mathcal{L}_{pk} + \sum_{l \in N_q} \mathcal{L}_{pl} \right), \tag{2}$$

explicitly includes the naturally available relationships from $p \in P'_q$ to $k \in P'_q \backslash p$ and $p$ to $N_q$, where $P'_q = P_q \cup \{q\}$. $P'_q$ implies the set of positives ($P_q$), including the query $q$. Similarly, $P'_q \backslash p$ denotes the set consisting of all positives and the query, except for the current sample that acts as a new query ($p$). $\mathcal{L}_{qk}$ and $\mathcal{L}_{pk}$ denote the objective functions that involve positive pairs, and $\mathcal{L}_{ql}$ and $\mathcal{L}_{pl}$ contain the negative pairs. Depending on the specific objective, these expressions are formulated to contrast with each other. The idea here is to convey to the model that the positives have a specific definition, with certain attributes in common, and that differ from the negatives. This enhances the neighbourhood structure of the embeddings. $\mathcal{L}_{anu-all}$ (Eq. 2) implies the framework that includes all possible additional relationships. We also investigate variants of ANU-ALL framework,

$$\mathcal{L}_{anu-h} = \mathcal{L}_{original} + \frac{1}{|Q|} \sum_{q \in Q} \sum_{p \in P_q} \left( \mathcal{L}_{pk'} + \mathcal{L}_{pl'} \right),$$
$$k' = \operatorname*{argmin}_{k \in P'_q \backslash p} \operatorname{sim}(p, k), \ l' = \operatorname*{argmax}_{l \in N_q} \operatorname{sim}(p, l), \ \text{i.e. (min pos sim \& max neg sim)} \tag{3}$$

$$\mathcal{L}_{anu-e} = \mathcal{L}_{anu-h},$$
$$k' = \operatorname*{argmax}_{k \in P'_q \backslash p} \operatorname{sim}(p, k), \ l' = \operatorname*{argmin}_{l \in N_q} \operatorname{sim}(p, l), \ \text{i.e. (max pos sim \& min neg sim)} \tag{4}$$

$\mathcal{L}_{anu-e}$ (Eq. 4) and $\mathcal{L}_{anu-h}$ (Eq. 24), which retains only the easiest and the hardest pairs from the newly introduced pairs. ANU-ALL variant $\mathcal{L}_{anu-all}$ (Eq. 2) is used by default in the experiments in this work unless specified otherwise. Appendix A.5 presents how our loss is related to others.

### 3.1 Multi-Similarity (MSim) Loss

For completeness, we briefly review the MSim Loss Wang et al. (2019).

#### 3.1.1 MSim Loss

The MSim loss Wang et al. (2019) proposes a general gradient-based pair-weighting framework to understand various pair-based metric learning algorithms. This identifies the absence of multiple similarities (relative and self-similarities) in the available objectives, and proposes the loss function

$$\mathcal{L}_{msim} = \frac{1}{|Q|} \sum_{q \in Q} \left\{ \frac{1}{\alpha} \log[1 + \sum_{k \in P_q} \exp(-\alpha(S_{qk} - \lambda))] \right.$$
$$\left. + \frac{1}{\beta} \log[1 + \sum_{l \in N_q} \exp(\beta(S_{ql} - \lambda))] \right\}, \tag{5}$$

where $S_{qk}$ is the dot-product similarity between the feature vectors $q$ and $k$ given by $\langle q, k \rangle$, $\alpha, \beta$, and $\lambda$ are empirically fixed hyperparameters.

The gradient of $\mathcal{L}_{msim}$ in Eq. 5, $w_{qv} = \left| \frac{\partial \mathcal{L}_{msim}}{\partial S_{qv}} \right|$ gives

$$w_{qv}^+ = \frac{\exp(-\alpha(S_{qv} - \lambda))}{1 + \sum_{k \in P_q} \exp(-\alpha(S_{qk} - \lambda))} \text{ and} \tag{6}$$

$$w_{qv}^- = \frac{\exp(\beta(S_{qv} - \lambda))}{1 + \sum_{l \in N_q} \exp(\beta(S_{ql} - \lambda))}, \tag{7}$$

where $w_{qv}^+$ is the weight associated with the positive pair $(x_q, x_v) \in P_q$ and $w_{qv}^-$ is the weight of a negative pair $(x_q, x_v) \in N_q$. These expressions incorporate self-similarities $(S_{qv})$ and relative similarities $(S_{ql} - S_{qv})$. Eqs. 5, 6, and 7 are borrowed from Wang et al. (2019). According to Wang et al. (2019), self-similarities alone are inadequate for precisely representing neighbourhood relationships in the latent space, as they influence optimization independently of adjacent pairs. To address this issue, relative similarities are introduced in the expression, which, together with self-similarities, can assist the model in better understanding the associations between these pairs.

However, these associations can be extended further. In the original loss expression in Eq. 5, we observe $S_{qk}$ and $S_{ql}$, which indicate the connections between query-positives and query-negatives alone. We hypothesize that, considering the other relationships between positives and assigning the query status to each positive in a training instance, the neighbourhood structure in the representation space could be maintained more accurately. We formalize this in a new loss function in the proposed ANU-RL framework as shown in Eq. 8.

### 3.1.2 MSim Loss in the Proposed Framework

The augmentation of the query in Eq. 5 can be seen in the proposed loss function in Eq. 8. It is to be noted that the change introduced in the proposed work still follows the gradient analysis performed in Wang et al. (2019).

$$\mathcal{L}_{anu-all-msim} = \frac{1}{|Q|} \sum_{q \in Q} \sum_{p \in P_q'} \left\{ \frac{1}{\alpha} \log[1 + \sum_{k \in P_q' \backslash p} \exp(-\alpha(S_{pk} - \lambda))] \right.$$
$$\left. + \frac{1}{\beta} \log[1 + \sum_{l \in N_q} \exp(\beta(S_{pl} - \lambda))] \right\}, \tag{8}$$

where the notations $Q, N_q, \alpha, \beta, P_q', S,$ and $\lambda$ follow from sections 3 and 3.1.1. $P_q' = \{q, p_1, p_2, p_3\}$ in Figure 1. The core idea of the proposed expression in Eq. 8 is illustrated in Figure 1, where the left-most part illustrates the mechanism of MSim loss Eq. 5, and the sketch in the middle illustrates the proposed push-pull mechanism. We can observe an additional loop over the set of positives, where in every looping instance, a positive sample $(p_i)$ from the comprehensive set of positives $(P_q')$ attracts the rest of the samples in the set $(P_q' \backslash p_i)$ and repels the negative samples $(N_q)$. Since the positives in themselves mean they are similar, hence they are shared in a common circle. Unlike this, negatives are self-contained in the inner circles, implying that they could be disjoint, as the same definition of the positive may not always apply to the negatives. Gradients computed over the aggregated distances in the additional summation introduced in Eq. 8 are used to update the model's parameters.

In a similar vein, we extend this to the following loss functions.

### 3.2 Triplet Loss

The aim of the Triplet loss Schroff et al. (2015) in Eq. 9 is to minimize the distance between images of the same identity while maximizing the distance between images of different identities.

$$\mathcal{L}_{triplet} = \sum_{q \in Q} \left[ d_{qp} + m - d_{qn} \right]_+, \tag{9}$$

$$\mathcal{L}_{anu-all-triplet} = \sum_{q \in Q} \sum_{p \in P'_q} \Big[ \sum_{k \in P'_q \setminus p} d_{kp} + m - d_{qn} \Big]_+, \tag{10}$$

where $P'_q = P_q \cup \{q\} = \{p, q\}$, $d_{ij}$ implies the distance between $i, j$ pair, and $m$ is a margin and is empirically fixed hyper-parameter. $[\cdot]_+$ implies a hinge function i.e. $\max(\cdot, 0)$.

### 3.3 FastAP Loss

Similar to Eq. 10, the proposed approach can be extended to the FastAP loss Cakir et al. (2019). Unlike most triplet-based losses, the FastAP loss Cakir et al. (2019) in Eq. 11 leverages the distance quantization technique for the distance list between the query and the reference embeddings. Likewise, this approximates the AP loss with binned histograms of distances to speed up the ranking.

$$\mathcal{L}_{fastAP} = \frac{1}{N_q^+} \sum_{j=1}^{L} \frac{H_j^+ h_j^+}{H_j}, \tag{11}$$

$$\mathcal{L}_{anu-all-fastAP} = \sum_{q \in Q} \sum_{\substack{p \in P'_q, \\ k = P'_q \setminus p}} \frac{1}{|k|} \sum_{j=1}^{L} \frac{H_{kj}^+ h_{kj}^+}{H_{kj}}, \tag{12}$$

where the details of the notations and the working principle of the loss function is presented in Appendix A.4.2 due to space constraints in the main paper.

### 3.4 An Analogy to Regularization

Expanding the $\mathcal{L}_{anu-all-msim}$ loss in Eq. 8 gives,

$$\begin{aligned}
\mathcal{L}_{anu-all-msim} &= \frac{1}{|Q|} \sum_{q \in Q} \frac{1}{\alpha} \log[1 + \sum_{k \in P_q} \exp(-\alpha(S_{qk} - \lambda))] + \frac{1}{\beta} \log[1 + \sum_{l \in N_q} \exp(\beta(S_{ql} - \lambda))] \\
&\quad + \frac{1}{|Q|} \sum_{q \in Q} \sum_{p \in P_q} \frac{1}{\alpha} \log[1 + \sum_{k \in P'_q \setminus p} \exp(-\alpha(S_{pk} - \lambda))] + \frac{1}{\beta} \log[1 + \sum_{l \in N_q} \exp(\beta(S_{pl} - \lambda))] \\
&= \mathcal{L}_{msim} + \mathcal{L}_{additional}
\end{aligned} \tag{13}$$

Further, this can be rewritten as

$$\mathcal{L}_{anu-all-msim} = \lambda_1 \times \mathcal{L}_{msim} + \lambda_2 \times \mathcal{L}_{additional}, \tag{14}$$

where $0 < \lambda_1, \lambda_2 \leq 1$ define the contribution of objectives $\mathcal{L}_{msim}$ and $\mathcal{L}_{additional}$. We experiment with a convex combination of these objectives, where both $\lambda_1$ and $\lambda_2$ can vary. In another case, we set $\lambda_1 = 1$ and vary $\lambda_2$ to study the contribution of the proposed constraints set $\mathcal{L}_{additional}$ alone. The latter setting mimics the standard regularization expression, where $\lambda_2$ acts as a regularization constant. The detailed analysis and the corresponding empirical results are provided in Appendix A.2.1.

Table 1: Details of various models used in this work

| Models | Backbone | Backbone + Agg Params | Trainable Params (M) | Feat Size |
|---|---|---|---|---|
| BoQ Ali-Bey et al. (2024) | ResNet50 He et al. (2016) | 8.5 + 10.0 (M) | 17.1 | 8192 |
| SALAD Izquierdo & Civera (2024) | DINOv2 Oquab et al. (2023) | 86.6 + 1.4 (M) | 88.0 | 8448 |
| MixVPR Ali-bey et al. (2023) | ResNet50 | 8.5 + 1.4 (M) | 8.5 | 4096 |
| R2F-G Zhu et al. (2023) | DeiT Touvron et al. (2021) | 21.9 (M) | 21.9 | 256 |
| NetVLAD Arandjelovic et al. (2016) | ResNet50 | 8.5 M + 32.8 K | 7.1 | 16384 |
| CosPlace Berton et al. (2022) | ResNet50 | 8.5 + 1.0 (M) | 8.1 | 1024 |
| ConvAP Ali-bey et al. (2022) | ResNet50 | 8.5 + 1.0 (M) | 8.1 | 4096 |
| SuperVLAD Lu et al. (2024c) | DINOv2 | 86.6 M + 3.8 K | 15.7 | 3072 |

Table 2: This presents a summary of the test datasets used in this work. Further details are provided in Appendix A.11

| Dataset | # Qry | # Db | Viewpoint | Illumination | Seasons | Temporal |
|---|---|---|---|---|---|---|
| P30k Torii et al. (2013) | 6.816K | 10K | ✓ | | | |
| Tokyo Torii et al. (2015) | 315 | 75.984K | ✓ | ✓ | | |
| Nordland Sünderhauf et al. (2013) | 27.592K | 27.592K | | | ✓ | |
| MSLS Warburg et al. (2020) | 740 | 18.871K | ✓ | ✓ | ✓ | |
| Amstertime Yildiz et al. (2022) | 1.23K | 1.23K | | | | ✓ |
| SPED Test Chen et al. (2018) | 607 | 607 | | ✓ | ✓ | |
| Eynsham Cummins & Newman (2009) | 23.935K | 23.935K | ✓ | | | |
| St Lucia Milford & Wyeth (2008) | 1.549K | 1.549K | ✓ | | | |
| SVOX Night Berton et al. (2021) | 823 | 17.166K | ✓ | ✓ | | |
| SVOX Overcast Berton et al. (2021) | 872 | 17.166K | ✓ | ✓ | | |
| SVOX Rain Berton et al. (2021) | 937 | 17.166K | ✓ | | ✓ | |
| SVOX Snow Berton et al. (2021) | 870 | 17.166K | ✓ | | ✓ | |
| SVOX Sun Berton et al. (2021) | 854 | 17.166K | ✓ | | ✓ | |

Table 3: Comparison between baseline (BL) MSim loss and its variants in the ANU-RL framework. None (BL) contains no additional pairs (ANU pairs). ALL variant includes all newly introduced ANU pairs. Hardest (AH) retains only the hardest pairs from ALL. Easiest (AE) variant, in contrast to Hardest, considers only the easiest pairs. We observe that the relatively high capacity models with high dimensional descriptors performing considerably better in AH or ALL case over the BL on most datasets. For example, BoQ, SALAD, MixVPR, and NetVLAD in AH, and R2Former, CosPlace, and ConvAP do well in ALL. SuperVLAD with small number of trainable parameters in the aggregator performs inconsistently across loss variants.

| Method | Dim | ANU pairs | Pittsburgh30k R@1 | R@5 | Tokyo 24/7 R@1 | R@5 | Nordland R@1 | R@5 | MSLS (Val) R@1 | R@5 | Amstertime R@1 | R@5 | SPEDTest R@1 | R@5 | Eynsham R@1 | R@5 |
|---|---|---|---|---|---|---|---|---|---|---|---|---|---|---|---|---|
| BoQ | 8192 | BL | 91.21 | 95.36 | 80.32 | 89.21 | 73.14 | 84.56 | 85.14 | 91.89 | 36.83 | 57.40 | 81.55 | 88.63 | 88.79 | 92.74 |
| | | ALL | 90.67 | 95.06 | 78.41 | 88.89 | 72.24 | 83.85 | 84.73 | 91.08 | 35.37 | 55.12 | 78.58 | 88.80 | 88.48 | 92.63 |
| | | AH | 91.51 | 95.67 | 84.44 | 90.79 | 78.88 | 88.19 | 86.62 | 91.49 | 38.37 | 58.46 | 82.70 | 91.76 | 89.56 | 93.27 |
| | | AE | 90.11 | 94.87 | 73.33 | 84.76 | 63.05 | 77.30 | 82.97 | 89.59 | 34.63 | 53.58 | 76.28 | 86.16 | 87.24 | 91.82 |
| SALAD | 8448 | BL | 92.06 | 96.16 | 91.75 | 97.14 | 78.88 | 89.78 | 91.22 | 95.41 | 52.28 | 74.15 | 90.44 | 95.88 | 90.53 | 94.48 |
| | | ALL | 91.86 | 96.16 | 93.33 | 96.83 | 78.66 | 88.88 | 90.00 | 95.54 | 53.01 | 73.66 | 90.28 | 94.56 | 90.55 | 94.63 |
| | | AH | 92.63 | 96.39 | 93.65 | 97.46 | 79.21 | 89.96 | 90.68 | 95.68 | 55.28 | 75.77 | 89.62 | 95.22 | 91.03 | 94.73 |
| | | AE | 90.68 | 95.32 | 87.62 | 94.92 | 67.55 | 80.18 | 89.86 | 94.73 | 47.24 | 69.11 | 87.64 | 93.90 | 90.06 | 94.30 |
| MixVPR | 4096 | BL | 90.35 | 95.10 | 79.37 | 89.21 | 75.18 | 86.31 | 83.65 | 90.27 | 35.20 | 54.07 | 84.68 | 93.74 | 88.03 | 92.20 |
| | | ALL | 90.61 | 95.25 | 75.87 | 89.52 | 75.37 | 86.73 | 84.32 | 90.54 | 37.48 | 55.85 | 81.22 | 91.10 | 88.26 | 92.29 |
| | | AH | 91.15 | 95.25 | 80.95 | 90.79 | 77.81 | 88.20 | 83.51 | 91.08 | 37.32 | 56.99 | 84.35 | 93.08 | 88.39 | 92.39 |
| | | AE | 89.69 | 95.04 | 70.79 | 85.08 | 59.79 | 75.29 | 83.24 | 89.86 | 34.15 | 53.01 | 76.61 | 88.47 | 87.11 | 91.75 |
| R2F | 256 | BL | 84.62 | 93.31 | 57.14 | 70.79 | 15.33 | 25.20 | 65.41 | 79.19 | 21.71 | 38.78 | 68.86 | 81.05 | 80.51 | 88.70 |
| | | ALL | 87.38 | 94.34 | 63.17 | 78.10 | 22.84 | 34.87 | 71.62 | 83.92 | 25.69 | 42.44 | 73.48 | 85.50 | 83.06 | 90.25 |
| | | AH | 86.22 | 93.79 | 58.10 | 76.19 | 21.18 | 33.12 | 69.19 | 82.16 | 23.82 | 41.38 | 72.98 | 84.84 | 82.64 | 90.00 |
| | | AE | 85.26 | 93.76 | 59.05 | 74.92 | 16.52 | 26.75 | 70.27 | 80.95 | 24.15 | 40.16 | 67.87 | 83.36 | 80.53 | 88.87 |
| NetVLAD | 16384 | BL | 89.99 | 94.97 | 69.84 | 81.27 | 68.45 | 81.97 | 82.16 | 88.92 | 30.73 | 49.67 | 79.74 | 89.95 | 87.23 | 91.84 |
| | | ALL | 89.86 | 94.63 | 71.43 | 82.22 | 69.18 | 82.24 | 82.16 | 89.19 | 30.41 | 48.94 | 76.11 | 87.64 | 87.67 | 92.29 |
| | | AH | 89.33 | 94.34 | 74.29 | 84.44 | 69.97 | 82.60 | 83.24 | 89.73 | 33.25 | 49.92 | 79.90 | 90.44 | 87.50 | 91.96 |
| | | AE | 88.31 | 94.04 | 62.86 | 73.02 | 46.15 | 61.64 | 80.14 | 87.03 | 25.53 | 42.85 | 71.83 | 85.01 | 86.96 | 91.93 |
| CosPlace | 1024 | BL | 89.10 | 94.51 | 66.98 | 79.05 | 60.25 | 75.59 | 80.00 | 89.97 | 29.51 | 47.64 | 79.57 | 89.79 | 87.42 | 92.02 |
| | | ALL | 89.22 | 94.60 | 68.89 | 83.17 | 64.36 | 78.97 | 82.03 | 89.46 | 29.35 | 48.78 | 78.58 | 88.80 | 87.28 | 91.73 |
| | | AH | 89.51 | 94.41 | 71.11 | 84.44 | 63.53 | 78.74 | 81.62 | 89.46 | 30.49 | 47.80 | 80.72 | 91.43 | 87.75 | 92.22 |
| | | AE | 88.54 | 94.18 | 60.63 | 75.56 | 48.85 | 65.78 | 79.46 | 88.24 | 27.56 | 45.04 | 74.46 | 84.84 | 85.90 | 91.11 |
| ConvAP | 4096 | BL | 89.69 | 95.14 | 76.83 | 85.08 | 63.33 | 77.65 | 76.22 | 85.14 | 33.41 | 51.46 | 81.88 | 92.26 | 86.17 | 91.06 |
| | | ALL | 90.23 | 95.29 | 75.56 | 85.40 | 65.50 | 79.48 | 80.54 | 88.38 | 34.72 | 52.03 | 80.56 | 89.95 | 86.45 | 91.33 |
| | | AH | 90.10 | 95.16 | 78.41 | 86.35 | 66.17 | 79.44 | 76.49 | 84.73 | 33.41 | 50.24 | 83.53 | 91.60 | 86.10 | 91.11 |
| | | AE | 89.41 | 94.72 | 63.17 | 76.83 | 53.53 | 70.84 | 78.24 | 86.08 | 30.08 | 47.24 | 75.78 | 88.63 | 84.72 | 90.35 |
| SupVLAD | 3072 | BL | 91.29 | 95.29 | 89.21 | 97.14 | 63.16 | 79.25 | 88.92 | 94.46 | 47.07 | 68.54 | 86.99 | 94.07 | 90.44 | 94.63 |
| | | ALL | 91.18 | 96.13 | 89.84 | 96.83 | 64.82 | 80.31 | 88.38 | 94.73 | 48.37 | 71.63 | 87.81 | 93.25 | 90.16 | 94.61 |
| | | AH | 91.46 | 96.17 | 89.84 | 96.19 | 61.86 | 78.36 | 88.78 | 94.46 | 45.37 | 68.13 | 87.48 | 93.25 | 90.03 | 94.44 |
| | | AE | 91.18 | 96.07 | 88.89 | 94.92 | 65.78 | 81.53 | 89.32 | 94.86 | 47.72 | 72.20 | 86.00 | 94.07 | 89.90 | 94.45 |

# 4 Experiments

This section presents the experimental setup, models, and datasets used in this study. A summary of the models and datasets are presented in Tables 1 and 2. Additional information is provided in Appendix in

Table 4: Notations and their definitions, and observations follow from Table 3. This presents the same analysis as in Table 3 but on different datasets.

| Method | Dim | ANU pairs | St Lucia | | SVOX | | SVOX Night | | SVOX Overcast | | SVOX Rain | | SVOX Snow | | SVOX Sun | |
|---|---|---|---|---|---|---|---|---|---|---|---|---|---|---|---|---|
| | | | R@1 | R@5 | R@1 | R@5 | R@1 | R@5 | R@1 | R@5 | R@1 | R@5 | R@1 | R@5 | R@1 | R@5 |
| BoQ | 8192 | BL | 60.66 | 86.48 | 97.99 | **98.99** | 58.57 | 74.97 | **96.56** | **98.28** | 91.89 | 97.01 | 96.21 | **98.85** | 87.24 | 95.43 |
| | | ALL | 60.59 | 86.41 | 97.69 | 98.79 | 60.27 | 75.94 | 96.22 | 97.94 | 89.22 | 96.37 | 94.37 | 97.93 | 83.02 | 91.45 |
| | | AH | **61.75** | **86.61** | **98.15** | 98.94 | **65.25** | **81.41** | 96.56 | 97.94 | 91.68 | 96.69 | **96.67** | 98.51 | **88.76** | **95.55** |
| | | AE | 61 | 86.41 | 97.11 | 98.57 | 50.67 | 68.41 | 94.95 | 97.36 | 85.38 | 93.92 | 93.79 | 98.05 | 79.86 | 90.05 |
| SALAD | 8448 | BL | **60.04** | 86.75 | **98.09** | 99.15 | 93.32 | 98.06 | **97.94** | 99.08 | 97.55 | 99.36 | 98.28 | **99.66** | 94.50 | 98.13 |
| | | ALL | 59.08 | 86.68 | 97.93 | 99.12 | 90.52 | 97.21 | 97.13 | 99.08 | **97.65** | **99.47** | 98.16 | 99.54 | 94.38 | 98.13 |
| | | AH | 59.29 | **86.95** | **98.24** | **99.26** | **94.29** | **98.66** | 97.71 | **99.31** | 97.44 | 99.47 | **98.85** | 99.66 | 94.50 | **98.48** |
| | | AE | 59.63 | 86.41 | 97.76 | 99.03 | 90.04 | 97.57 | 97.59 | 98.85 | 97.33 | 99.04 | 97.59 | 99.43 | **94.73** | 98.36 |
| MixVPR | 4096 | BL | 61.41 | 86.20 | 96.62 | 98.28 | 44.59 | 63.79 | 93.23 | 97.13 | 86.45 | 93.38 | 92.53 | 97.24 | 77.87 | 86.77 |
| | | ALL | 61.07 | **86.41** | 96.81 | **98.40** | 52.00 | 70.84 | 93.69 | 97.59 | 88.58 | 94.77 | 93.56 | 98.05 | 79.63 | 89.58 |
| | | AH | 60.66 | 86.13 | **97.07** | 98.38 | 50.91 | 69.02 | **94.27** | 97.48 | 88.05 | 94.45 | 92.64 | 98.05 | **81.15** | **90.52** |
| | | AE | **61.68** | 86.00 | 96.30 | 98.19 | 44.71 | 62.21 | 92.43 | 96.10 | 84.74 | 92.10 | 91.49 | 97.01 | 72.37 | 85.71 |
| R2F | 256 | BL | 50.61 | 78.21 | 88.65 | 94.66 | 7.41 | 17.01 | 74.31 | 85.09 | 50.16 | 66.81 | 55.63 | 74.83 | 40.98 | 57.61 |
| | | ALL | **53.28** | **82.17** | **91.90** | **96.25** | **12.39** | **24.79** | **80.50** | **90.71** | **62.75** | **80.58** | **67.01** | **83.45** | **50.70** | **65.46** |
| | | AH | 52.39 | 80.60 | 91.11 | 95.90 | 9.60 | 20.78 | 76.38 | 88.99 | 58.38 | 75.56 | 61.49 | 78.51 | 44.03 | 60.30 |
| | | AE | 51.98 | 81.42 | 89.79 | 95.14 | 9.72 | 20.90 | 75.69 | 87.04 | 61.26 | 75.67 | 64.37 | 81.49 | 47.66 | 63.70 |
| NetVLAD | 16384 | BL | 60.66 | **86.48** | 97.53 | 98.78 | 45.44 | 68.41 | 94.61 | 97.36 | 90.07 | 95.41 | 93.68 | **97.93** | 83.72 | **92.97** |
| | | ALL | **61.68** | 86.27 | 97.21 | 98.73 | 44.11 | 65.98 | 94.27 | 97.48 | 88.37 | 95.52 | 94.37 | 97.93 | 79.63 | 90.40 |
| | | AH | 60.38 | 86.13 | 97.31 | 98.74 | 50.79 | 67.56 | 93.46 | 97.59 | 86.98 | 93.81 | 92.41 | 97.36 | 80.68 | 90.16 |
| | | AE | 59.77 | 85.93 | 96.72 | 98.31 | 32.93 | 55.04 | 94.50 | 98.28 | 84.10 | 93.49 | 92.53 | 97.24 | 72.95 | 84.78 |
| CosPlace | 1024 | BL | 59.90 | 85.66 | 96.57 | 98.28 | 32.44 | 50.67 | 92.09 | 97.02 | 80.26 | 90.50 | 90.34 | **97.36** | 69.67 | 83.26 |
| | | ALL | 59.77 | 85.93 | **96.79** | **98.42** | **40.22** | **59.78** | **93.81** | **97.25** | **84.53** | **92.10** | 92.41 | 97.24 | **74.24** | **85.95** |
| | | AH | 60.45 | **86.07** | **96.69** | 98.49 | 36.33 | 55.89 | 92.43 | 96.10 | 82.39 | 91.14 | 91.38 | 96.78 | 72.01 | 82.44 |
| | | AE | **61.68** | 85.72 | 96.14 | 98.17 | 36.82 | 55.53 | 92.66 | 96.33 | 82.07 | 91.36 | 90.00 | 96.90 | 67.33 | 82.08 |
| ConvAP | 4096 | BL | 58.95 | 85.25 | 94.99 | 97.09 | 19.32 | 33.90 | 81.88 | 91.63 | 69.26 | 80.68 | 77.13 | 88.51 | 58.31 | 74.71 |
| | | ALL | **60.66** | **86.00** | 95.16 | **97.35** | **32.44** | **48.60** | **88.42** | **94.50** | **77.27** | 86.02 | **86.55** | **94.71** | **67.56** | **80.44** |
| | | AH | 60.38 | 85.59 | **95.22** | 97.29 | 20.53 | 33.17 | 82.91 | 90.14 | 68.94 | 80.79 | 76.78 | 88.28 | 60.30 | 76.35 |
| | | AE | 59.90 | 85.04 | 93.98 | 96.75 | 26.85 | 46.05 | 87.27 | 93.35 | 76.63 | **88.79** | 85.06 | 94.25 | 62.41 | 78.69 |
| SupVLAD | 3072 | BL | 58.74 | 86.41 | **97.53** | **99.00** | 82.75 | 92.71 | 96.90 | 98.13 | 95.41 | 99.36 | 96.09 | 99.08 | 94.26 | 98.13 |
| | | ALL | **59.22** | **86.82** | 97.11 | 98.76 | 85.78 | 94.05 | **97.02** | 98.51 | 95.09 | 98.93 | 95.75 | **99.43** | 93.68 | **98.59** |
| | | AH | 58.67 | 86.61 | 97.17 | 98.85 | 80.92 | 92.10 | 96.10 | 98.51 | 92.32 | 98.51 | 95.75 | 99.20 | 91.80 | 97.42 |
| | | AE | 58.54 | 86.41 | 97.13 | 98.83 | **85.78** | **94.78** | 96.90 | 98.62 | **95.62** | 98.72 | 95.86 | 99.20 | 93.91 | 98.48 |

Table 5: This table compares the recalls of NetVLAD-WPCA+Ours and Patch-NetVLAD+Ours with the recalls of NetVLAD-WPCA+MSim and Patch-NetVLAD+MSim. Patch-NetVLAD-P refers to the performance version of the Patch-NetVLAD, and Patch-NetVLAD-S refers to the speed version. We see that the widely used models in the proposed ANU-RL framework showing improved R@1 and R@5 performance over BL in most cases.

| Method | Dim | Pittsburgh30k | | Tokyo 24/7 | |
|---|---|---|---|---|---|
| | | R@1 | R@5 | R@1 | R@5 |
| NetVLAD-WPCA+MSim | 4096 | 89.06 | 94.38 | **76.83** | 83.81 |
| **NetVLAD-WPCA+ALL** | 4096 | **89.74** | **94.73** | 75.56 | **85.08** |
| Patch-NetVLAD-S+MSim | 100x4096 | 82.95 | 92.56 | 51.75 | 72.70 |
| **Patch-NetVLAD-S+ALL** | 100x4096 | **83.07** | **92.90** | **57.78** | **78.41** |
| Patch-NetVLAD-P+MSim | 100x4096 | 84.38 | 93.03 | 59.05 | 79.68 |
| **Patch-NetVLAD-P+ALL** | 100x4096 | **84.77** | **93.19** | **63.49** | **80.95** |

section A.11. In this work, we integrate the proposed framework with MSim Wang et al. (2019), Triplet Schroff et al. (2015), and FastAP Cakir et al. (2019) losses. However, it is a plug-and-play approach that can be integrated with any contrastive loss function, including N-Pairs loss Schroff et al. (2015), Generalized Lifted Structure loss Hermans et al. (2017), and others. The experiments are conducted in two stages: 1) training the VPR models with the baseline losses (aggregator+BL), and 2) training within the proposed ANU-RL framework (**aggregator+ANU**). In both settings, we use the same training configuration to ensure a fair comparison.

To investigate the impact of the triplet hardness/informativeness of the additional triplets on the model performance, we modify the ANU-RL framework in three different ways and run the following experiments: 1. **None (BL)**: The baseline approach, 2. ANU-**ALL**: The full framework, which incorporates all additional relationships, 3. ANU-**Hardest**: Includes only the hardest ANU positive (a positive pair with the lowest

similarity) and negative (a negative pair with the highest similarity) pairs, and 4. ANU-**Easiest**: The opposite of case 3, using only the ANU easiest positive (with the highest positive similarity) and negative (with the lowest negative similarity) pairs. The bold part indicates how we denote them in tables. To avoid long recurring definitions, we adopt the shorthand BL for None, i.e., baseline, ALL for ANU-ALL, AH for ANU-Hardest, and AE for ANU-Easiest, in the following discussion. References to the "lowest" and "highest" similarities are local, meaning they apply only within the ANU triplets (the newly introduced pairs) and not across the entire dataset. These variants are then evaluated across a broad suite of VPR datasets that encompass a variety of real-world challenges. Additionally, we perform a qualitative analysis using t-SNE distribution visualizations and loss curvature plots of the models.

All experiments use the ANU-ALL variant as the default, unless otherwise stated. The proposed variants apply to loss functions that operate on multiple positive and negative examples per query. In contrast, a few loss functions, such as the triplet loss, involve exactly one positive and one negative sample per query. In this case, only the ANU-ALL variant is applicable.

### 4.1 Implementation

The proposed framework is developed in PyTorch. Apart from this, the rest of the experiments in this work follow the MixVPR setting Ali-bey et al. (2023). The training and evaluation scripts used for most aggregators, including Conv-AP Ali-bey et al. (2022), CosPlace Berton et al. (2022), and MixVPR aggregators, are borrowed from the public GitHub repository [1]. Except for R2Former-GR Zhu et al. (2023), SALAD Izquierdo & Civera (2024), and SuperVLAD Lu et al. (2024c), the rest of the models use the ResNet50 He et al. (2016) cropped at its last layer as the backbone. We use only the global retrieval component of R2Former (R2Former-GR or R2F), ignoring the reranker. The input images are resized to 224×224 for training and evaluation.

### 4.2 Training and Evaluation

We use the GSV-Cities dataset Ali-bey et al. (2022) for training the models. We use a batch size of 100, and the Stochastic Gradient Descent (SGD) optimizer with an initial learning rate of 0.025. All models are trained for 40 epochs. For evaluation, the recall@k (R@k) metric is used. We report R@1 and R@5 results. The trained models are evaluated on the popular benchmark datasets (In Table 2 and more on this in Appendix (A.11)). For the evaluation, we follow the Patch-NetVLAD inference scripts.

## 5 Discussion

This section compares VPR aggregators trained using the original MSim loss with those trained under the proposed framework. The study focuses exclusively on single-stage VPR models. The discussion is structured as follows.

### 5.1 Quantitative Analysis

Tables 3, 4, and 5 present the recall@k comparison of various VPR techniques in the proposed framework against baseline models. The results of a few baseline models reproduced here are slightly lower than the off-the-shelf results in the published papers. This could be due to smaller input resolution, a smaller batch size, and changing other related hyper-parameters in our work. These are made to match the available computational budget. The Nordland dataset in this work is taken from Hausler et al. (2021), where images of the tunnel scene are removed. We see from Tables 3, 4, and 5 that the State Of The Art (SOTA) aggregators, BoQ, SALAD, MixVPR, and NetVLAD showing improvement over their BL counterparts in the case of AH. ALL and AH cases are assumed to be relatively highly informative than the other cases due to inclusion of the harder pairs. Particularly, the BoQ and SALAD models consistently show an improved performance on the most challenging datasets including, Pittsburgh 30k with severe viewpoint variations, Nordland with extreme appearance changes and visual aliasing, Amstertime with significant viewpoint and appearance changes, and SVOX Night with large day-night shifts.

---

[1] https://github.com/amaralibey/MixVPR.git

We discuss some of the highlights of the models trained within our framework. BoQ in AH improves over BoQ+BL by ∼4 points from 80.32 to 84.44 on Tokyo 24/7, ∼6 points from 73.14 to 78.88 on Nordland, ∼2 points from 36.83 to 38.37 on Amstertime, around 7 points from 58.57 to 65.25 on the SVOX Night datasets, etc. These are a few representative examples. Nevertheless, BoQ+AH outperforms BoQ+BL in R@1 on almost all datasets, with only a few exceptions where the performance drop is negligible. Similarly, another recent work, SALAD, in AH improves over BL by around 2 points on most datasets. We see a drop in performance by SALAD+AH on some of the datasets, such as MSLS (Val) and SPED Test. However, those cases are minimal. Likewise, MixVPR drops in R@1 on SPEDTest, St Lucia, and a few others, where the drop is smaller compared to gains, such as 44.59 to 50.91 on SVOX Night like extremely challenging datasets. We observe that the widely used NetVLAD aggregator in AH, outperforms the BL by more than 5 points on the complex SVOX Night dataset. In contrast, it is slightly sensitive to variations in images due to rain, snow, and sun, showing a minor drop in R@1 on the variants of the SVOX dataset. On the other hand, R2F in ALL case, consistently surpasses BL by a large margin, and in the AH case, it offers the second best performance on majority of the datasets. CosPlace+ALL achieves a minimum of around 2 points and a maximum of around 8 points gain in R@1 over BL on the variants of the SVOX dataset in Table 4. Similarly, the ConvAP+ALL aggregator achieves a minimum gain of 7 points and a maximum of 13 points over BL on challenging SVOX variants. The best performance of ConvAP and CosPlace models show a slight inconsistency between AH and All cases, and R2F consistently do well in ALL case. One reason could be due to relatively low-dimensional features, which maybe less capable of learning from the hardest pairs. In contrast, the SuperVLAD aggregator is inconsistent across the datasets, where the number of datasets, on which the model in each of the ANU-RL variants perform the best is almost uniformly distributed. This makes it hard to recommend the best performing ANU-RL variant for SuperVLAD.

A few aggregators like SuperVLAD shows better performance in the AE case on some datasets. This can be attributed to insufficient capacity of a model to be able to learn from the hard pairs. From Table 1, we see a small number of trainable parameters the SuperVLAD aggregator contains, which could be one reason for the inconsistent performance.

Additionally, we compare NetVLAD-WPCA to the two-stage ranking technique Patch-NetVLAD in Table 5. The NetVLAD reported in Table 5 uses WPCA (NetVLAD-WPCA) to reduce the dimensionality of the descriptors to 4096. We use only a single patch size of 5 for Patch-NetVLAD evaluation. We notice that the proposed loss improves the NetVLAD-WPCA, Patch-NetVLAD-S (Speed version: RSS matcher), and Patch-NetVLAD-P (Performance version: RANSAC matcher) consistently on Pittsburgh 30k and Tokyo24x7 datasets.

The improvements are attributable to ANU-RL, as the training setup is identical except for the loss function. Moreover, the proposed framework incurs no additional computational or storage cost at inference.

## 5.2 Qualitative Analysis

This section presents t-SNE visualizations and top@1 predictions. Due to space constraints, limited results are discussed here. Extended results are presented in Appendix in section A.6. This analysis uses the MSim loss.

Figure 3 illustrates the t-SNE Van der Maaten & Hinton (2008) feature distributions of the SOTA VPR models BoQ and SALAD. These plots visualize only 20 query-database pairs for convenience in understanding. The query and its top-1 prediction share the label (0-19) and are denoted by different colors (green for the query and blue for the database samples). For an easy catch, a few pairs are circled, where the proposed approach separates them better than the BL. We observe from Figure 3 (b) that the proposed approach, BoQ+AH, pushes the query-prediction closer and pulls it sufficiently far from the rest of the pairs. On the other hand, the BoQ+BL in Figure 3 (a) does it relatively poorly. Similarly, the SALAD+AH in Figure 3 (d) does a better job than the SALAD+BL in Figure 3 (c). While there are a few pairs where both models appear to separate them poorly, the proposed method makes fewer mistakes than the baseline losses, as demonstrated across various experiments.

Additionally, Figure 2 shows the top@1 predictions of SOTA VPR models on challenging datasets. These include significant changes in viewpoint, illumination, and scale. We see from Figure 2 that, although there

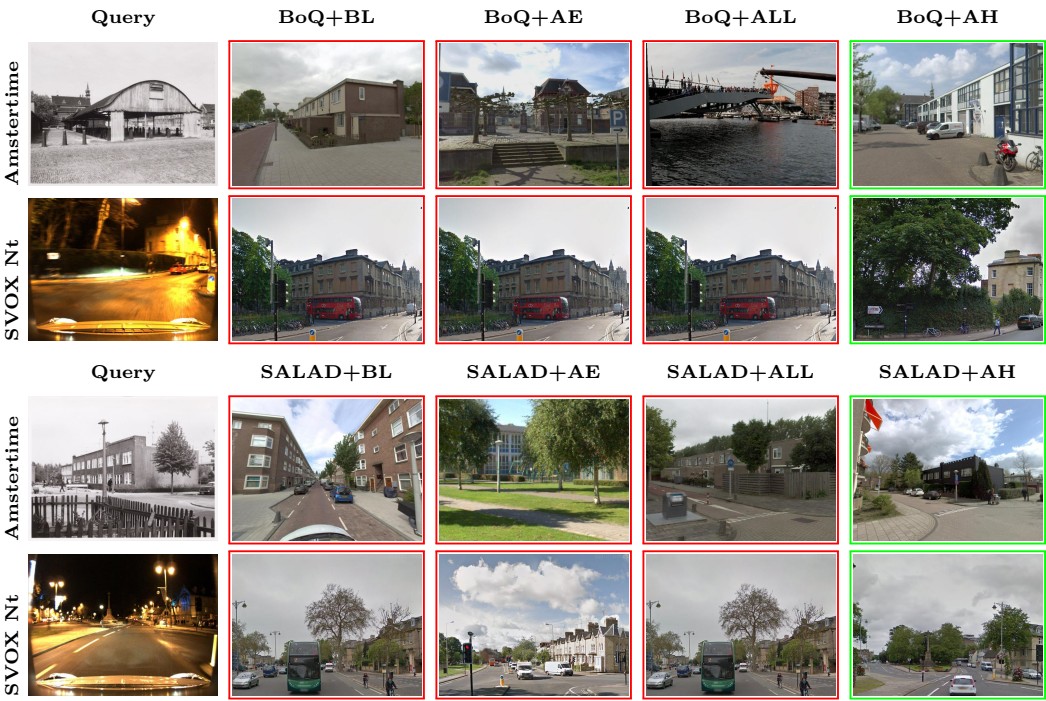

Figure 2: This figure shows the top@1 retrieval outcomes of the VPR models across various scenarios. This is with MSim (Eq. 5) loss and Ours (Eq. 8) loss in AE, ALL, and AH variants. A Red bounding box denotes incorrect predictions, while a Green bounding box indicates correct predictions. These are a few representative top@1 retrievals by the BoQ and the SALAD models from Amstertime and SVOX Nt datasets. Despite the significant appearance, illumination, and viewpoint variations, we see that the models trained with the MSim loss in the proposed AH variant retrieving the correct reference images. This implies that the proposed approach is helping the model to capture the subtle details shared between query and reference images, which is essential for feature discriminability.

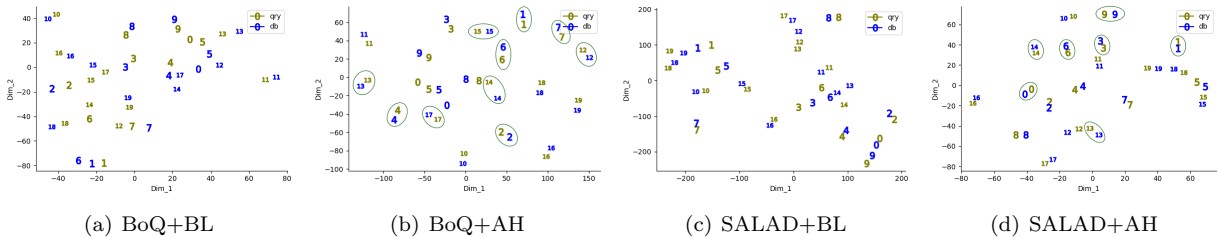

Figure 3: t-SNE plots illustrating the feature separability. This is with the original MSim ($\mathcal{L}_{msim}$) and MSim in AH variant of the ANU-RL framework. These visualizations show the better separation between the pairs in (b) and (d) by the proposed framework compared to that of the baseline approaches in (a) and (c). These plots are obtained for the Amstertime dataset, which contains large appearance and viewpoint changes between query and reference galleries. Annotated pairs highlight the better separation achieved by our approach.

are other closely matching reference images, the other approaches are fooled by the confusing patterns, while our method (AH) correctly retrieves. This implies that the additional information that the proposed framework injects into the MSim loss helps it capture subtle similarities between corresponding pairs.

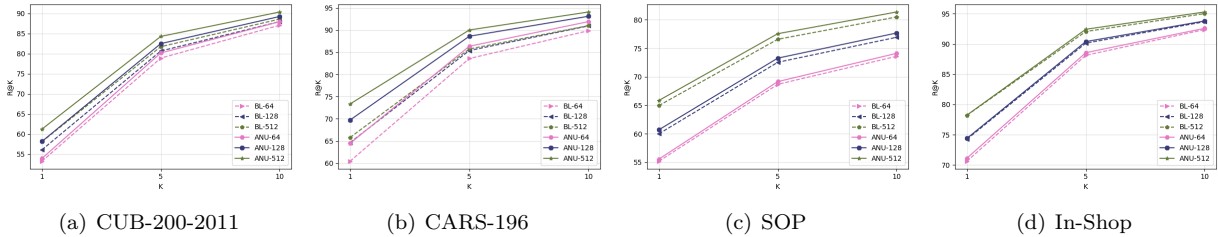

| (a) CUB-200-2011 | (b) CARS-196 | (c) SOP | (d) In-Shop |

Figure 4: Comparing the original triplet loss against triplet loss in the ANU-RL framework. We run this on popular image retrieval datasets, including CUB-200-2011, CARS-196, SOP, and In-Shop. The presented recalls for each feature dimension (64, 128, 512) are averaged over five runs. These experiments follow MSim implementation. We observe that the triplet loss in the proposed framework outperforms the original triplet loss consistently on all datasets.

## 6 Utility Beyond VPR and Future Research

Visual representation learning is fundamental to numerous vision tasks, such as image recognition and classification. The core concept behind these algorithms is to learn compact representations that capture the complex relationships among the training samples. Our proposed framework is not limited to a specific application, such as visual place recognition. This is a plug-and-play approach that can be applied to any representation learning algorithm.

To study the generalizability of the proposed framework, we extend it to the broader Image Retrieval (IR) problem. We use Triplet and MSim losses for this study, where the IR model and its implementation follow Wang et al. (2019). With MSim loss, either a drop or a gain in performance is minor and unnoticeable. Therefore, we exclude MSim experiments for IR. In contrast, triplet loss in our framework surpasses the original approach across almost all popular datasets consistently with a large margin. This is illustrated in Figure 4.

As a future study, we will extend this to various other losses, such as SupCon Khosla et al. (2020), N-Pair Sohn (2016), and Lifted-Structure Oh Song et al. (2016), to name a few. In our future work, we will also develop an API that allows users to integrate any contrastive loss into our framework easily.

## 7 Limitations

Computation of similarity scores between the additional terms introduced by ANU-RL, slightly increases the training latency. With optimal implementation, this could be reduced. However, with the Triplet loss, the increase in the training cost is almost negligible.

## 8 Conclusion

This work introduced a simple and logical framework that is plug-and-play and can be applied to any contrastive loss. The proposed approach contains an additional loop over the set of positives of each query in a training instance to ensure a better approximation of the neighbourhood relationships in the latent space. Specifically, we applied the proposed framework to the Multi-Similarity loss and Triplet loss, which are popular in VPR, and the FastAP loss. We demonstrated overall better performance over the widely used aggregators on the challenging benchmark datasets, including Pittsburgh 30k, Tokyo 24/7, Nordland, MSLS (Val), and many other datasets with significant real-time variations. Importantly, the improvements achieved with the proposed approach over these aggregators introduce no additional computational overhead and storage demands at test time.

**Broader Impact Statement** Due to space constraints, this section is moved to Appendix A.1.

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

# A Appendix

**Contents**

**Note that the experiments in Appendix follow ANU-ALL variant by default unless specified otherwise.**

### A.1 Broader Impact Statement

The proposed method, while demonstrated to improve the performance of VPR/geo-localisation in this work, is not limited to VPR. Our framework can be integrated with any representation learning (RL) algorithm to enhance its accuracy at zero inference cost. Its potential benefits and limitations are discussed below.

**Positive Impact**: Our approach is both conceptually simple and easy to implement. Moreover, it incurs no computational overhead during inference. As such, it can positively impact a wide range of pair-based objectives in representation learning, which underpins many deep learning algorithms. From the VPR perspective, this approach enhances localization accuracy in autonomous navigation of mobile robots, particularly in GPS-denied environments. Additionally, the proposed framework improves various applications of image retrieval such as surveillance.

**Negative Impact**: In certain cases, we see inconsistent performance of some of the aggregators in the proposed framework. Given that VPR is often deployed in risk-sensitive applications such as autonomous driving and navigation, understanding its failure modes is crucial. Unanticipated failures could lead to significant consequences. Therefore, we are yet to have a complete theoretical explanation for cases in which our approach degrades the accuracy of specific algorithms.

Further, in addition to autonomous navigation, the VPR and image retrieval can also be used for applications such as surveillance, tracking, and privacy-sensitive localization. Unethical use of the proposed approach in these sensitive applications should be avoided. In particular, in localization, the VPR is not a stand-alone approach; rather, it is integrated with other components that can help prevent catastrophic failures in safety-critical applications.

In practice, as stated, reasoning-based approaches help understand the cause of the failure. To achieve this, the recent trend turned towards MLLM-based methods. However, this comes at the cost of increased complexity during training and testing.

**Response**   The potential misuse of the approach in the mentioned sensitive applications is not just limited to our approach. It applies to other models as well. Nevertheless, unethical use the proposed approach should be avoided.

### A.2 Interpreting the MSim loss in the Proposed Framework

#### A.2.1 An Analogy to Regularization

We present a brief theoretical analysis of the proposed framework. Particularly, we use Multi-Similarity (MSim) loss for this analysis. The MSim loss in the proposed framework is given by

$$\mathcal{L}_{msim-ours} = \frac{1}{|Q|} \sum_{q \in Q} \sum_{p \in P'_q} \left\{ \frac{1}{\alpha} \log[1 + \sum_{k \in P'_q \setminus p} \exp(-\alpha(S_{pk} - \lambda))] + \frac{1}{\beta} \log[1 + \sum_{l \in N_q} \exp(\beta(S_{pl} - \lambda))] \right\}. \quad (15)$$

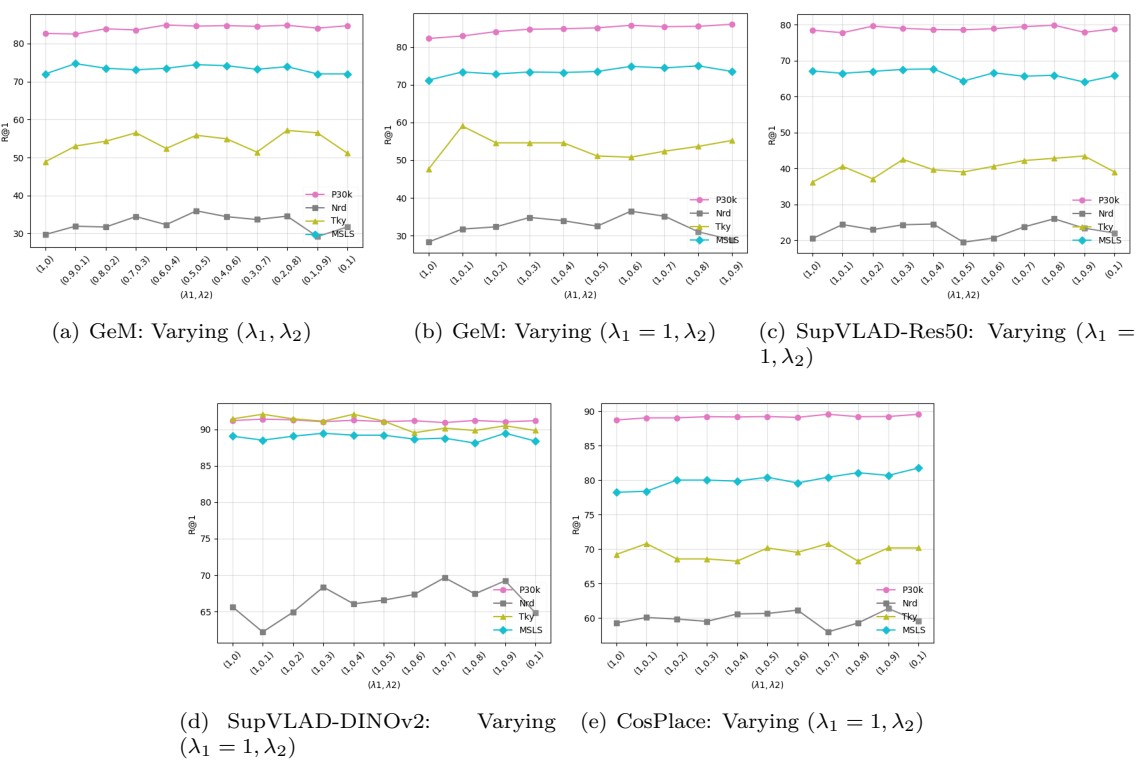

(a) GeM: Varying $(\lambda_1, \lambda_2)$

(b) GeM: Varying $(\lambda_1 = 1, \lambda_2)$

(c) SupVLAD-Res50: Varying $(\lambda_1 = 1, \lambda_2)$

(d) SupVLAD-DINOv2: Varying $(\lambda_1 = 1, \lambda_2)$

(e) CosPlace: Varying $(\lambda_1 = 1, \lambda_2)$

Figure 5: Contribution of different objectives ($\mathcal{L}_{msim}$ and $\mathcal{L}_{additional}$ as in Sec. A.2.1) in the proposed framework that is the regularization setup. $(\lambda_1, \lambda_2)$ (a) We see that while decreasing the importance of the MSim loss and simultaneously increasing that for the proposed constraint set, almost across all weights and on all datasets, we see a rise in the overall performance over the MSim alone (1,0) case. (b) This is when we maintain the contribution of the MSim loss and vary the contribution of the additional constraints proposed in this work; we observe a similar behaviour, where the performance improvement is consistent. (a) and (b) are with the GeM model, with the number of samples per place being 8. The remaining models use 4 samples per place.

Expanding this further gives the following equation,

$$
\mathcal{L}_{anu-all-msim} = \frac{1}{|Q|} \sum_{q \in Q} \frac{1}{\alpha} \log[1 + \sum_{k \in P_q} \exp(-\alpha(S_{qk} - \lambda))] + \frac{1}{\beta} \log[1 + \sum_{l \in N_q} \exp(\beta(S_{ql} - \lambda))]
$$

$$
+ \frac{1}{|Q|} \sum_{q \in Q} \sum_{p \in P_q} \left( \frac{1}{\alpha} \log[1 + \sum_{k \in P'_q \setminus p} \exp(-\alpha(S_{pk} - \lambda))] + \frac{1}{\beta} \log[1 + \sum_{l \in N_q} \exp(\beta(S_{pl} - \lambda))] \right)
$$

$$
= \mathcal{L}_{msim} + \mathcal{L}_{additional}
$$

$$
\mathcal{L}_{additional} = \frac{1}{|Q|} \sum_{q \in Q} \sum_{p \in P_q} \frac{1}{\alpha} \log[1 + \sum_{k \in P'_q \setminus p} \exp(-\alpha(S_{pk} - \lambda))] + \frac{1}{|Q|} \sum_{q \in Q} \sum_{p \in P_q} \frac{1}{\beta} \log[1 + \sum_{l \in N_q} \exp(\beta(S_{pl} - \lambda))]
$$

$$
= \frac{1}{|Q|} \frac{1}{\alpha} \log \left( \prod_{q \in Q} \prod_{p \in P_q} \left[ 1 + \sum_{k \in P'_q \setminus p} \exp(-\alpha(S_{pk} - \lambda)) \right] \right) + \frac{1}{|Q|} \frac{1}{\beta} \log \left( \prod_{q \in Q} \prod_{p \in P_q} \left[ 1 + \sum_{l \in N_q} \exp(\beta(S_{pl} - \lambda)) \right] \right)
$$

$$
= a \log x + b \log y.
$$

(16)

Empirical analysis related to $\mathcal{L}_{msim}$ and $\mathcal{L}_{additional}$ is shown in Figure 5.

### A.2.2 Informativeness of the Proposed Additional Constraints

$$\mathcal{L}_{additional} = \frac{1}{|Q|} \sum_{q \in Q} \sum_{p \in P_q} \frac{1}{\alpha} \log[1 + \sum_{k \in P'_q \setminus p} \exp(-\alpha(S_{pk} - \lambda))] + \frac{1}{|Q|} \sum_{q \in Q} \sum_{p \in P_q} \frac{1}{\beta} \log[1 + \sum_{l \in N_q} \exp(\beta(S_{pl} - \lambda))].$$

(17)

Since the terms inside the logarithm are non-negative, we get

$$\log[1 + \sum_{k \in P'_q \setminus p} \exp(-\alpha(S_{pk} - \lambda))] \geq \log[\sum_{k \in P'_q \setminus p} \exp(-\alpha(S_{pk} - \lambda))],$$

$$\log[1 + \sum_{l \in N_q} \exp(\beta(S_{pl} - \lambda))] \geq \log[\sum_{l \in N_q} \exp(\beta(S_{pl} - \lambda))].$$

(18)

Further, using the LogSumExp, the smooth approximation of the max function gives

$$\max(\{-\alpha(S_{pk} - \lambda)\}_k) \approx \log[\sum_{k \in P'_q \setminus p} \exp(-\alpha(S_{pk} - \lambda))],$$

$$\max(\{\beta(S_{pl} - \lambda)\}_k) \approx \log[\sum_{l \in N_q} \exp(\beta(S_{pl} - \lambda))].$$

(19)

This implies that weighting the most informative pairs is taken into account inherently. However, this is a smooth weighting, unlike hard mining, which discards the uninformative pairs with hard constraints.

### A.2.3 Gradient Analysis of the MSim Loss

$$\mathcal{L}_{msim} = \frac{1}{|Q|} \sum_{q \in Q} \left\{ \frac{1}{\alpha} \log[1 + \sum_{k \in P_q} \exp(-\alpha(S_{qk} - \lambda))] \right.$$

$$+ \frac{1}{\beta} \log[1 + \sum_{l \in N_q} \exp(\beta(S_{ql} - \lambda))] \right\}$$

$$\frac{\partial \mathcal{L}_{msim}}{\partial \theta} = \frac{1}{|Q|} \sum_{q \in Q} \left\{ -\sum_{k \in P_q} \frac{\exp((-\alpha(S_{qk} - \lambda)))}{[1 + \sum_{k \in P_q} \exp(-\alpha(S_{qk} - \lambda))]} \frac{\partial S_{qk}}{\partial \theta} \right.$$

$$+ \sum_{l \in N_q} \frac{\exp(\beta(S_{ql} - \lambda))}{[1 + \sum_{l \in N_q} \exp(\beta(S_{ql} - \lambda))]} \frac{\partial S_{ql}}{\partial \theta} \right\}$$

(20)

### A.2.4 Gradient Analysis of the MSim Loss in the Proposed Framework

$$\mathcal{L}_{anu-all-msim} = \frac{1}{|Q|} \sum_{q \in Q} \sum_{p \in P'_q} \left\{ \frac{1}{\alpha} \log[1 + \sum_{k \in P'_q \setminus p} \exp(-\alpha(S_{pk} - \lambda))] \right.$$

$$+ \frac{1}{\beta} \log[1 + \sum_{l \in N_q} \exp(\beta(S_{pl} - \lambda))] \right\}$$

$$\frac{\partial \mathcal{L}_{anu-all-msim}}{\partial \theta} = \frac{1}{|Q|} \sum_{q \in Q} \sum_{p \in P'_q} \left\{ -\sum_{k \in P'_q} \frac{\exp((-\alpha(S_{pk} - \lambda)))}{[1 + \sum_{k \in P'_q} \exp(-\alpha(S_{pk} - \lambda))]} \frac{\partial S_{pk}}{\partial \theta} \right.$$

$$+ \sum_{l \in N_q} \frac{\exp(\beta(S_{pl} - \lambda))}{[1 + \sum_{l \in N_q} \exp(\beta(S_{pl} - \lambda))]} \frac{\partial S_{pl}}{\partial \theta} \right\}$$

(21)

Unlike standard L1 and L2 regularizers, the parameters do not appear explicitly in the above expressions; instead, they are present through log-exponential functions. This indirect involvement, both in the additional constraints in Sec. A.2.1 and in the gradient computations in Sec. A.2.4, makes the analysis inconclusive. Further investigation is needed to support the empirical performance reported in this work.

### A.3 Space and Time Complexity Analysis

Table 6 compares training time per epoch, cuda memory for training, and R@1 performance of different VPR models trained with the original MSim loss and the loss in ANU-RL framework. This is with MSim loss. We notice that the proposed approach takes more training time per epoch. On the other hand, our approach improves performance. However, the trade-off between performance and complexity is natural. The performance numbers presented here are computed for the Pittsburgh 30k dataset on a workstation with an NVIDIA GeForce RTX 3090 GPU with 24 GB of memory.

| | Dims | MSim | | | MSim-Ours | | | Triplet | | | Triplet-Ours | | |
|---|---|---|---|---|---|---|---|---|---|---|---|---|---|
| | | GPU (GB)/BS | TT/E (min) | R@5 (%) | GPU (GB)/BS | TT/E (min) | R@5 (%) | GPU (GB)/BS | TT/E (min) | R@5 (%) | GPU (GB)/BS | TT/E (min) | R@5 (%) |
| BoQ | 8192 | 9.5/400 | 8 | 95.36 | 9.5/400 | 13 | $95.06_{(-0.3)}$ | 9.5/400 | 9 | 92.71 | 9.5/400 | 10 | $92.96_{(+0.25)}$ |
| SALAD | | 23/240 | 11 | 96.16 | 23/240 | 15 | $96.16_{(0)}$ | 23/240 | 13 | 94.31 | 23/240 | 13 | $94.64_{(+0.33)}$ |
| MixVPR | 4096 | 9/400 | 3.5 | 95.10 | 9/400 | 7.3 | $95.25_{(+0.15)}$ | 9/400 | 3.75 | 92.83 | 9/400 | 3.75 | $93.13_{(+0.3)}$ |
| R2Former-GR | 256 | 23/400 | 6 | 93.31 | 23/400 | 10 | $94.34_{(+1.03)}$ | 23/400 | 7 | 90.83 | 23/400 | 7 | $92.08_{(+1.25)}$ |
| CosPlace | 1024 | 10/400 | 3.62 | 94.51 | 10/400 | 7.57 | $94.60_{(+0.09)}$ | 10/400 | 3.53 | 92.50 | 10/400 | 3.58 | $92.18_{(-0.32)}$ |
| ConvAP | 4096 | 6.5/400 | 3.38 | 95.14 | 6.5/400 | 7.47 | $95.29_{(+0.15)}$ | 6.5/400 | 3.45 | 91.39 | 6.5/400 | 3.48 | $91.97_{(+0.58)}$ |
| SuperVLAD | 3072 | 18/400 | 8 | 95.29 | 18/400 | 12.3 | $96.13_{(+0.84)}$ | 18/400 | 8 | - | 18/400 | 8 | - |

Table 6: This table compares the GPU memory, time taken per epoch, and performance of different methods with MSim ($\mathcal{L}_{msim}$) and Ours ($\mathcal{L}_{anu-all-msim}$), the ANU-ALL variant. TT/E: Training Time/Epoch. BS: Batch Size.

The complexity analysis is performed entirely on the NVIDIA RTX 3090 GPU. However, to speed up the experiments, we split them across different machines, including the RTX 3090 and L40S. The MixVPR, BoQ, SuperVLAD, R2F-G, and SALAD were run on RTX 3090, while the other models were run on the L40S GPU.

Specifically, in our implementation, we follow a sequential approach by looping over the positives for each query; hence, the memory complexity remains the same. In addition, time complexity scales as O(N), where N is the number of positives for a query ($P_q$) or the number of images sampled from each city. Compared to the BL, we repeat the loss computation for $P_q$ times; hence, it is O(N).

### A.4 Extending the ANU-RL Framework to Triplet and FastAP Losses Applied to VPR

Tables 8 and 9 compare performance of VPR models trained with original Triplet and FastAP loss functions against their modified versions in our framework. We notice that the VPR models trained with the proposed loss variants surpasses the performance of the baseline models by a large margin. This implies that the proposed framework generalizes very well to multiple metric learning functions.

| Dim | CUB-200-2011 | | | | CARS-196 | | | | SOP | | | |
|---|---|---|---|---|---|---|---|---|---|---|---|---|
| | MSim | | ANU-MSim | | MSim | | ANU-MSim | | MSim | | ANU-MSim | |
| | R@1 | R@5 | R@1 | R@5 | R@1 | R@5 | R@1 | R@5 | R@1 | R@5 | R@1 | R@5 |
| 64 | 57.26 | 82.42 | **57.3** | **82.51** | 66.75 | **87.99** | **65.85** | 87.51 | 59.78 | 72.87 | **60.03** | **72.95** |
| 128 | 61.49 | **84.81** | **61.66** | 84.55 | **73.91** | **91.13** | 72.07 | 90.25 | 65.15 | 77.14 | **65.32** | **77.20** |
| 512 | 66.09 | 87.25 | **66.45** | **87.34** | **80.79** | **94.08** | 80.21 | 93.69 | 71.39 | 82.31 | **72.53** | **83.13** |

Table 7: All experiments are average over 5 runs. The proposed framework is particularly helpful in cases of domain shift such as Tokyo 24/7, Nordland, and SVOX Night datasets with severe appearance variation. In other cases, the gains are small, as we see in this table.

#### A.4.1 Multi-Sim Loss for Image Retrieval

Table 7 presents Multi-Sim results on the image retrieval application that are excluded from the main paper. We see the inconsistency in performance gains with the ALL case. In particular, on the Cars-196 dataset, our approach performs slightly worse. From empirical investigation, we observe that the proposed variants are primarily helpful in challenging scenarios such as domain shifts. For instance, with the BoQ+AH, we see performance gains of around 3 points on the Nordland dataset with extreme seasonal variations, 5 points on the Tokyo 24/7 dataset with severe illumination changes, 2 points on the Amstertime dataset with large

appearance changes due to time gaps, and 6 points on the SVOX-Night dataset with significant day-night shifts.

### A.4.2 FastAP Loss

In general, Average Precision (AP) is used as an evaluation metric that we typically aim to maximize. However, instead of limiting it to evaluation, recent research has tried to use AP directly as the training objective. The fastAP loss is one such approach. However, AP in its original form is non-differentiable due to the discrete ranking that involves an indicator function. To make it smooth and enable meaningful gradient computation for feature learning, the fastAP loss removes sorting entirely and introduces a distance-quantization technique. In other words, the precision and recall in AP (Eq. 22),

$$\text{AP} = \int_{\Omega} \text{Prec}(z)\, d\,\text{Rec}(z), \tag{22}$$

are approximated with smooth functions of the distance between query and database embeddings. $z \in \Omega$ denotes the continuous distance values between query and database feature representations. At a high level, these distances are uniformly divided into bins, and the database images are assigned to the corresponding bins based on distance scores. Upon solving Eq. 22 using distance distributions, we obtain the final fastAP expression in Eq. (9) of the main paper, which approximates the AP. In the final expression, $H_j^+$ is the cumulative positive retrievals until the bin $j$, $h_j^+$ denotes the number of positive retrievals in the $j^{th}$ bin, $H_j$ computes the total cumulative retrievals until the $j^{th}$ bin, and $N_q^+$ denotes the total number of positives for the query $q$. In addition, our approach incorporates the flipped roles of the query and positives, resulting in Eq. (10) in the main paper.

### A.5 Relation with Existing Loss Functions

The evolution of contrastive loss functions has been built on small changes rather than on something entirely new. Some widely used loss functions are broadly related as follows.

$$\begin{aligned} \mathcal{L}_{\text{contrast}} \subseteq \mathcal{L}_{\text{triplet}} \subseteq \mathcal{L}_{in-trip-mine} \subseteq \mathcal{L}_{lsl} \subseteq \mathcal{L}_{n-pair} \subseteq \\ \mathcal{L}_{GLS-BA} \subseteq \mathcal{L}_{sup-con} \subseteq \mathcal{L}_{ms} \subseteq \mathcal{L}_{ANUs-RL-ms}. \end{aligned} \tag{23}$$

The early contrastive loss, $\mathcal{L}_{\text{contrast}}$, operates with pairs, positives, and negatives in different updates independently. The subsequent work, $\mathcal{L}_{\text{triplet}}$ loss, combines both positives and negative pairs, and optimizes the loss with both in a single update; hence, it could handle the problem that $\mathcal{L}_{\text{contrast}}$ tries to address. Further, In-triplet mining, $\mathcal{L}_{in-trip-mine}$, introduces an additional distance $d_{pn}$ connecting positive to negative. The final expression includes either $d_{an}$ or $d_{pn}$, whichever is harder. Effectively dealing with triplets, which could match with the triplet loss if the $d_{pn}$ is less informative. For example, during the cross-seasons scenario, say the Nordland dataset, positive and negative images come from the same season, summer, while the query is from winter. In that case, $d_{an}$ is harder than $d_{pn}$, reducing the loss to the triplet loss. The Lifted-structure loss, $\mathcal{L}_{lsl-hard}$, the harder version of it, is closely related to the $\mathcal{L}_{in-trip-mine}$ loss. The difference is that the $\mathcal{L}_{lsl-hard}$ samples the hardest pairs within the batch. The smoother version of it, $\mathcal{L}_{lsl-smooth}$, includes all pair-wise connections, including positive to negatives. This inherently solves the problem that $\mathcal{L}_{in-trip-mine}$ tries to address. In contrast, the N-Pairs loss, $\mathcal{L}_{n-pair}$, extends the triplet variants to incorporate multiple negatives. The Generalized Lifted Structure loss (GLS) with the Batch-All case, $\mathcal{L}_{GLS-BA}$, extends the triplet loss, which proposes an efficient pair mining strategy, exploiting same-class pairs as positives and samples from all other classes as negatives. The subsequent work, Supervised Contrastive Loss, $\mathcal{L}_{sup-con}$, does the same job but with different expressions. Similarly, a contemporary work with SupCon loss, the Multi-Similarity loss, $\mathcal{L}_{ms}$, closely follows the $\mathcal{L}_{GLS-BA}$ loss with a minor difference. Broadly, we can classify these loss functions based on the relationships they use as follows. 1. Single positive and negative pairs, 2. single positive and multiple negatives, 3. multiple positives and multiple negatives, and 4. the proposed approach introduces a new class with all possible inter-connections between them. If we observe closely, these works evolved with minor improvements over the preceding ones.

Table 8: Performance comparison of VPR models on the challenging datasets with the Triplet loss Schroff et al. (2015). agg+Trip implies the aggregator trained with the original Triplet loss in (9) and **agg+ALL** refers to the aggregator trained with the modified Triplet loss within ANU-RL framework in (10). We observe that the proposed framework outperforming the original loss function in majority of the instances.

| Method | Pittsburgh30k | | Tokyo 24/7 | | Nordland | | MSLS (Val) | |
|---|---|---|---|---|---|---|---|---|
| | R@1 | R@5 | R@1 | R@5 | R@1 | R@5 | R@1 | R@5 |
| NetVLAD+Trip | 80.52 | 90.62 | 32.38 | 52.38 | 8.57 | 15.03 | 67.30 | 78.51 |
| **NetVLAD+ALL** | **80.66** | **90.74** | **38.41** | **56.83** | **10.77** | **18.64** | **68.24** | **78.78** |
| ConvAP+Trip | 80.58 | 91.39 | 29.21 | 44.13 | 8.95 | 16.19 | 61.22 | 75.54 |
| **ConvAP+ALL** | **83** | **91.97** | **32.38** | **51.75** | **13.6** | **23.88** | **65.81** | **77.43** |
| CosPlace+Trip | **84.02** | **92.50** | 38.41 | 54.92 | 21 | 35.86 | 70.68 | 80.68 |
| **CosPlace+ALL** | 83.80 | 92.18 | **43.17** | **58.10** | **22.94** | **38.08** | 70.68 | **81.08** |
| MixVPR+Trip | **85.05** | 92.83 | 45.71 | **63.49** | 24.05 | 39.80 | 70.95 | 82.16 |
| **MixVPR+ALL** | 84.90 | **93.13** | **46.03** | 60.63 | **24.46** | **40.66** | **73.38** | **82.43** |
| R2Former-GR+Trip | 78.89 | 90.83 | 42.54 | 61.90 | 9.62 | 16.95 | 56.22 | 75.27 |
| **R2Former-GR+ALL** | **81** | **92.08** | **46.67** | **62.86** | **13.08** | **22.50** | **62.16** | **78.65** |
| SALAD+Trip | 87.78 | 94.31 | 82.22 | **91.43** | **61.71** | **77.14** | 85 | **93.38** |
| **SALAD+ALL** | **87.94** | **94.64** | **83.17** | 90.79 | 55.87 | 70.03 | **87.16** | **93.38** |
| Boq+Trip | 85.14 | 92.71 | **44.44** | 60.95 | 26.22 | 42.77 | 74.73 | 84.46 |
| **BoQ+ALL** | **86.14** | **92.96** | 42.86 | **64.13** | **32.87** | **50.47** | **75.41** | **84.73** |
| SuperVLAD+Trip | 87.81 | 94.66 | **78.10** | 88.89 | 36.45 | 51.05 | 84.19 | **93.92** |
| **SuperVLAD+ALL** | **88.31** | **94.84** | 77.14 | 88.89 | **42.06** | **57.28** | **84.32** | 93.38 |

Table 9: Performance comparison of VPR models on the challenging datasets with FastAP loss Cakir et al. (2019). agg+FastAP implies the aggregator trained with the baseline FastAP loss (11) and **agg+ALL** refers to the aggregator trained with the modified FastAP loss (12) within the ANU-RL framework. Like with the other loss functions, we notice a similar trend with this loss as well, where models in our framework consistently improves upon BL in majority of the tests.

| Method | Pittsburgh30k | | Tokyo 24/7 | | Nordland | | MSLS (Val) | |
|---|---|---|---|---|---|---|---|---|
| | R@1 | R@5 | R@1 | R@5 | R@1 | R@5 | R@1 | R@5 |
| NetVLAD+FastAP | 85.96 | 93.16 | 49.52 | 64.76 | **31.57** | **46.26** | **76.76** | **84.59** |
| **NetVLAD+ALL** | **86.12** | **93.47** | **52.06** | **66.03** | 28.10 | 42.66 | 75 | 84.32 |
| ConvAP+FastAP | 86.55 | 93.85 | 43.81 | 60.95 | 30.65 | 48.01 | **72.30** | 82.03 |
| **ConvAP+ALL** | **87.79** | **93.98** | **56.51** | **71.43** | **42.91** | **60.91** | **72.30** | **83.24** |
| CosPlace+FastAP | 85.78 | 92.94 | 50.48 | 65.71 | 25.83 | 41.62 | 72.84 | **84.19** |
| **CosPlace+ALL** | **86.62** | **93.43** | **56.83** | **70.48** | **33.10** | **49.50** | **75.68** | 84.05 |
| MixVPR+FastAP | 87.02 | 93.90 | 53.02 | 67.62 | 40.56 | **58.84** | 76.35 | 85.14 |
| **MixVPR+ALL** | **87.56** | **93.96** | **55.87** | **72.70** | 41 | 58.22 | **77.57** | **85.41** |
| R2Former-GR+FastAP | 82.09 | 92.17 | 52.06 | 69.84 | 9.13 | 15.63 | 59.59 | 74.59 |
| **R2Former-GR+ALL** | **83.41** | **92.65** | **52.70** | **71.43** | **11.67** | **19.57** | **63.24** | **76.62** |
| SALAD+FastAP | 89.51 | **94.87** | 81.27 | 90.16 | 60.09 | 74.18 | 86.89 | **94.73** |
| **SALAD+ALL** | **89.64** | 94.85 | **87.30** | **93.33** | **61.53** | **75.87** | **87.30** | 93.51 |

## A.6 Extended Qualitative Results

This section discusses visualizations of the t-SNE plots and top@1 predictions.

### A.6.1 t-SNE Visualization

Figure 6 shows the t-SNE distributions of VPR models trained with the original MSim loss and with our loss. Some of the data point pairs are highlighted with annotations. The green annotation implies the best separability by the proposed approach, while all other do this poorly. On the other hand, the red mark denotes the poor separation by our approach, where at least one of the rest of the approaches does this better.

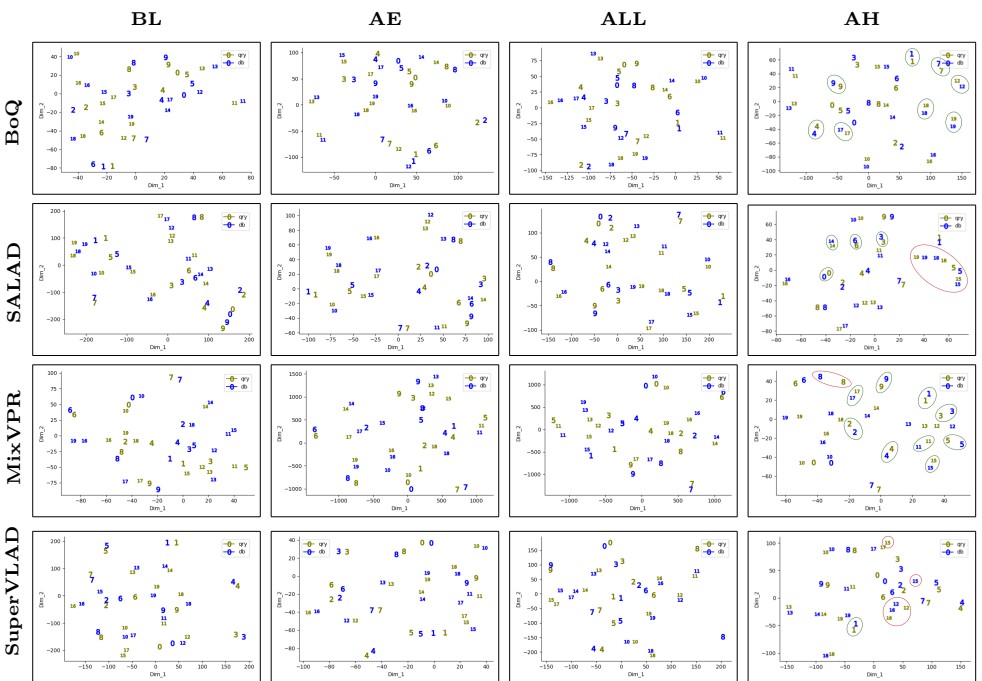

Figure 6: t-SNE plots illustrating the feature separability. Red annotations indicate poor separability by AH and at least of one of the rest of the cases do better for the pair. Green annotation indicate the best separation for the pair over all the rest of the cases

### A.6.2   Top@1 Retrievals

Top@1 retrievals of the VPR models in various challenging scenarios are presented in Figs. 7, 8, 9, 10, and 11.

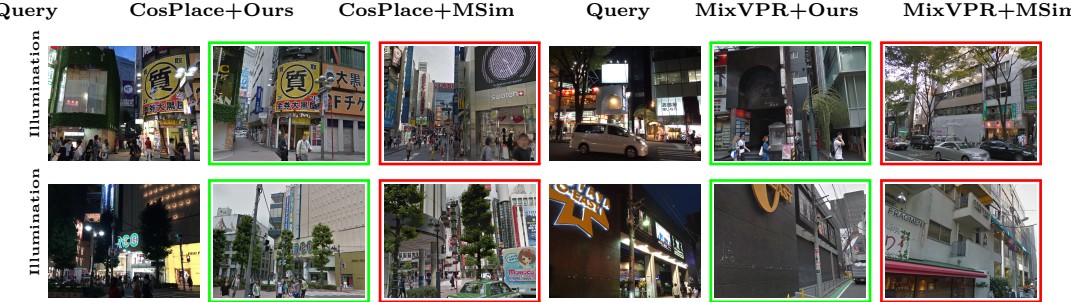

Figure 7: This figure shows the top@1 retrieval of the VPR models across illumination variation scenarios. This is with MSim ($\mathcal{L}_{msim}$) and Ours ($\mathcal{L}_{anu-all-msim}$) losses. A Red bounding box denotes incorrect predictions, while a Green bounding box indicates correct predictions. We see the models trained with MSim getting confused with false matching structures at night light and retrieving an incorrect reference image. On the other hand, our method correctly retrieves in all the challenging cases. These examples are from the Tokyo 24x7 dataset.

### A.7   Normalization or Scale Match

Following the regularization setting, in all of our experiments, we set $\lambda_i$ in Eq. 14 to 1. Moreover, most losses with the same hyperparameters often vary in scale. In our work, the Multi-Sim, Triplet, and FastAP loss functions show different loss scales. Therefore, we do not see it as necessary but rather as a means of

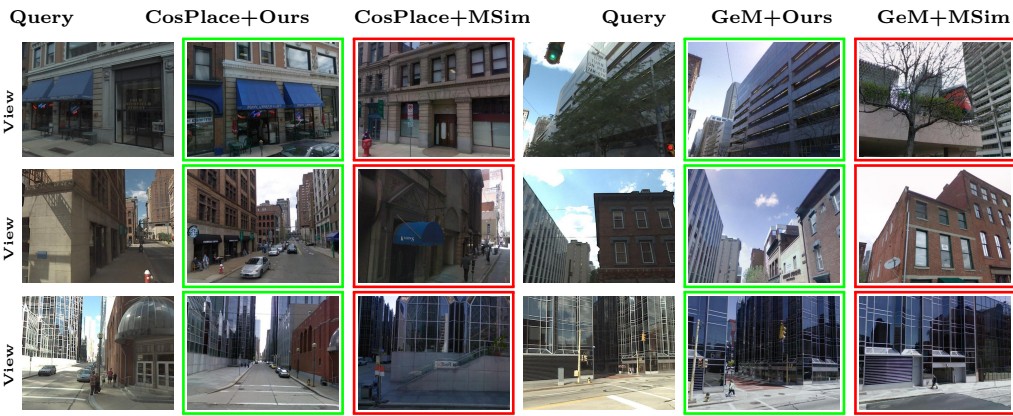

Figure 8: The loss used and the bounding boxes follow from Figure 7. This is a representative example of a viewpoint varying scenario. Despite having the perfect match for the query in the reference database, we see MSim models based on the partial matches arrive at an incorrect location. Our approach predicts the correct matches for all the queries in this example. These examples are taken from the Pittsburgh 30k dataset.

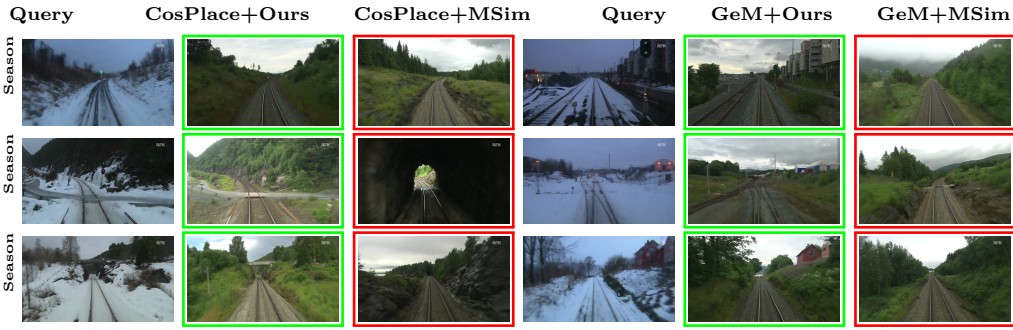

Figure 9: The loss used and the bounding boxes follow from Figure 7. This is an interesting example from the Nordland dataset, where the pairs contain perceptual aliasing and seasonal variation. Due to the railway track being dominant in the images, it's hard to locate the query in the reference correctly. This is because the visual difference between any image pair is negligibly small, although they are geographically far apart. For example, it is hard to identify the matching and unmatching pairs in the triplet, top right, and bottom right. Similarly, in the triplets in the right half, the visual differences in the structures are located to the sides of the track. However, the MSim might be failing to pay attention to them and retrieve incorrect matches. Ours works well.

maintaining numerical stability, which the proposed losses do not run into. Regardless of this, we present results with a few representative models with the proposed losses ANU-ALL and ANU-H normalized in Tables 10 and 11. We see a drop in most cases compared to un-normalized cases. This could be because the normalization resets the strength of the additional terms for optimization. Moreover, we view the proposed framework as a regularization technique, in which, as in standard approaches, we introduce $\lambda$, setting it to 1 in all our experiments. When we normalize the loss ALL and AH cases, we are picking one of the possible values of the $\lambda$. The best $\lambda$ in standard techniques like L1/L2 regularizers is often selected empirically. Similarly, in our experiments, we used $\lambda = 1$. Nevertheless, future research can sweep over a range of $\lambda$ values and select the best one based on the downstream application.

Furthermore, when we compare the training loss curves for the MSim loss and our losses, we observe no significant scale difference between them. The baseline training begins with a loss magnitude of 1, and ours begins with 2.5, for the same set of hyper-params, which is in the acceptable margin.

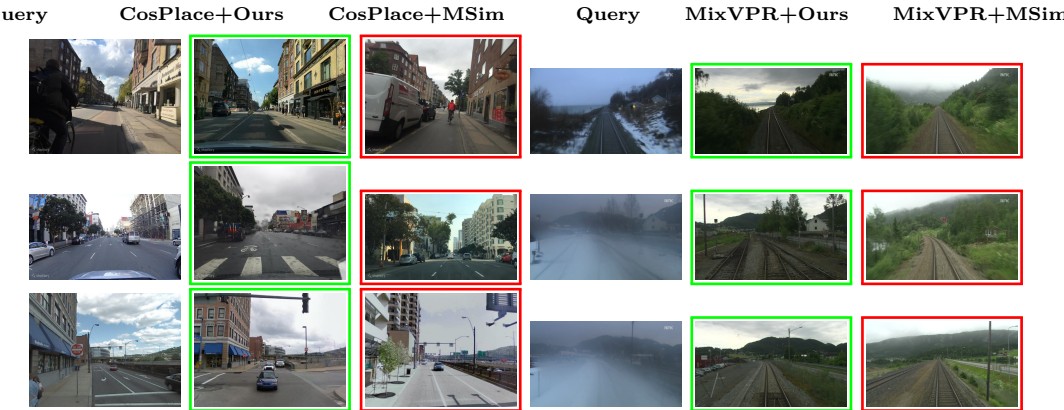

Figure 10: The loss used and the bounding boxes follow from Figure 7. The right half of the triplets is taken from the MSLS dataset and depicts the scale changes. When the query images are zoomed out, the scene covers additional patterns in the query that are absent in the reference images. Due to this, the MSim model falsely matches the query-reference images. The right half of the triplets depicts the occlusion caused by haze in the query images and seasonal variation between the pairs. Our approach is robust to these variations and retrieves the right matches.

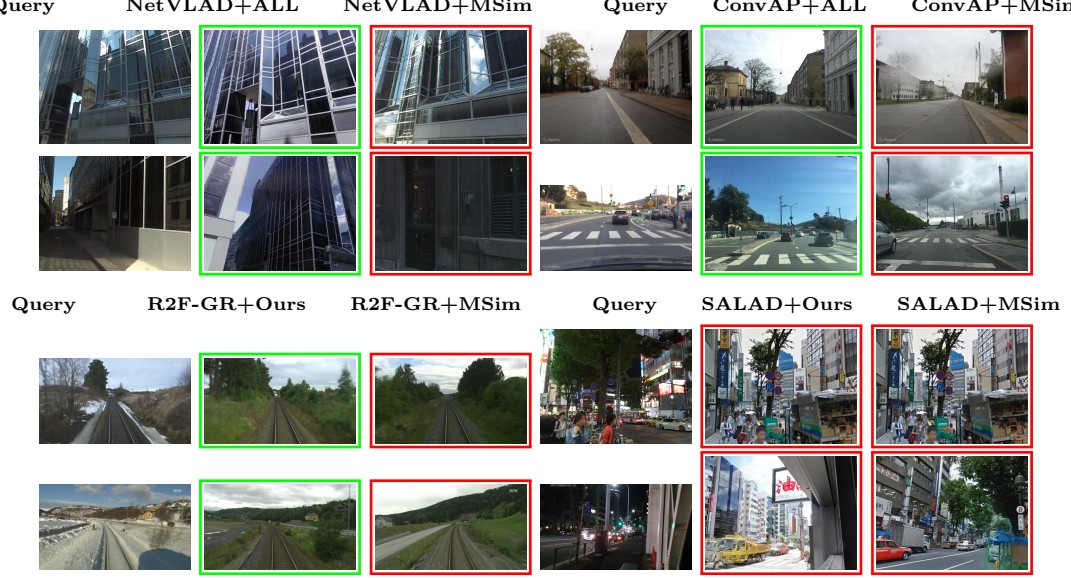

Figure 11: The loss used and the bounding boxes follow from Figure 14. The scenarios depicted include repetitive patterns in the leftmost columns of the top row of images, blur and illumination variations in the rightmost columns of the top row, perceptual aliasing in the leftmost columns of the bottom row, and cluttered scenes in the rightmost columns of the bottom row. Our method correctly retrieves in all the challenging cases, except in cluttered examples. R2F: R2Former.

## A.8 Selection Protocol of the ANU-RL Variant

In the majority of the cases, ANU-ALL and ANU-Hardest both outperform the baseline approaches. AH (AE) involves only the hardest (the easiest) similarity term, while ALL involves all multiple similarities. Therefore, AH/AE involves slightly reduced computation; hence, we recommend AH, the best-performing variant among AH and AE.

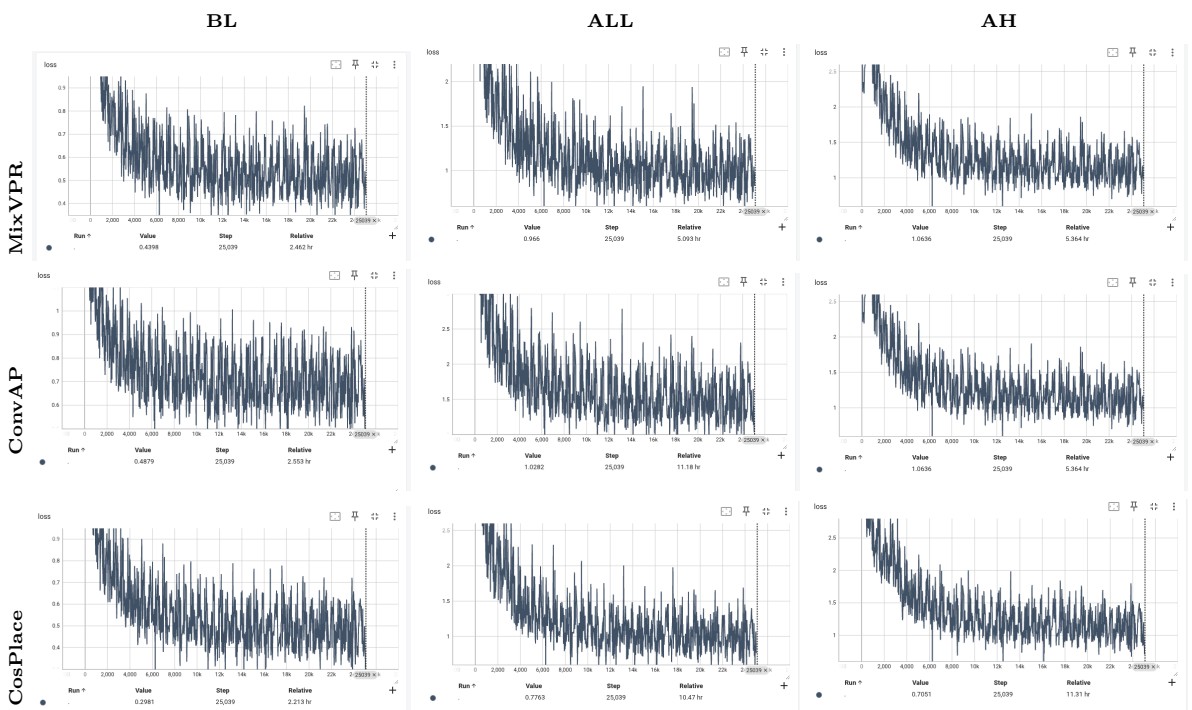

Figure 12: Training plots.

Table 10: Performance analysis with normalization of the proposed ANU-RL variants. We notice a drop in performance in most cases with the proposed variants. Scaling the loss function is similar to scaling the regularization term. The drop in performance in this case could be because of the wrong choice of the $\lambda$ in our framework.

| Method | Dim | ANU pairs | Pittsburgh30k | | Tokyo 24/7 | | Nordland | | MSLS (Val) | | Amstertime | | SPEDTest | | Eynsham | |
|---|---|---|---|---|---|---|---|---|---|---|---|---|---|---|---|---|
| | | | R@1 | R@5 | R@1 | R@5 | R@1 | R@5 | R@1 | R@5 | R@1 | R@5 | R@1 | R@5 | R@1 | R@5 |
| BoQ | 8192 | BL | 91.21 | 95.36 | 80.32 | 89.21 | 73.14 | **84.56** | 85.14 | **91.89** | 36.83 | **57.40** | 81.55 | 88.63 | 88.79 | 92.74 |
| | | ALL | 90.67 | 95.06 | 78.41 | 88.89 | 72.24 | 83.85 | 84.73 | 91.08 | 35.37 | 55.12 | 78.58 | 88.80 | 88.48 | 92.63 |
| | | AH | 91.51 | 95.67 | 84.44 | 90.79 | 78.88 | 88.19 | 86.62 | 91.49 | 38.37 | 58.46 | 82.70 | 91.76 | 89.56 | 93.27 |
| | | ALL-Norm | 90.55 | 94.92 | 80.95 | 87.94 | 67.66 | 81.10 | 84.86 | 90.68 | 35.85 | 54.96 | 78.75 | 87.81 | 87.86 | 92.17 |
| | | AH-Norm | **91.53** | **95.82** | **81.59** | **90.16** | **73.54** | 84.41 | **85.54** | 90.27 | **38.94** | 57.24 | **82.04** | **91.27** | **89.29** | **93.18** |
| MixVPR | 4096 | BL | **90.35** | 95.10 | 79.37 | 89.21 | **75.18** | **86.31** | **83.65** | 90.27 | 35.20 | 54.07 | **84.68** | **93.74** | **88.03** | 92.20 |
| | | ALL | 90.61 | 95.25 | 75.87 | 89.52 | 75.37 | 86.73 | 84.32 | 90.54 | 37.48 | 55.85 | 81.22 | 91.10 | 88.26 | 92.29 |
| | | AH | 91.15 | 95.25 | 80.95 | 90.79 | 77.81 | 88.20 | 83.51 | 91.08 | 37.32 | 56.99 | 84.35 | 93.08 | 88.39 | 92.39 |
| | | ALL-Norm | 90.17 | **95.19** | 79.05 | **90.16** | 72.41 | 84.47 | 82.97 | 89.05 | 35.37 | **55.45** | 83.53 | 92.09 | 87.79 | 92.05 |
| | | AH-Norm | 90.07 | 94.89 | **79.68** | 89.52 | 72.72 | 84.59 | 81.89 | 89.32 | **36.18** | 53.17 | **84.68** | 93.08 | 87.94 | **92.22** |
| CosPlace | | BL | **89.10** | 94.51 | 66.98 | 79.05 | **60.25** | **75.59** | 80 | **89.97** | 29.51 | 47.64 | **79.57** | 89.79 | 87.42 | 92.02 |
| | | ALL | 89.22 | 94.60 | 68.89 | 83.17 | 64.36 | 78.97 | 82.03 | 89.46 | 29.35 | 48.78 | 78.58 | 88.80 | 87.28 | 91.73 |
| | | AH | 89.51 | 94.41 | 71.11 | 84.44 | 63.53 | 78.74 | 81.62 | 89.46 | 30.49 | 47.80 | 80.72 | 91.43 | 87.75 | 92.22 |
| | | ALL-Norm | 88.89 | 94.41 | 68.89 | 80.63 | 58.59 | 74.38 | 80.27 | 86.62 | **30.73** | **49.43** | 77.59 | 88.30 | 87.09 | 91.81 |
| | | AH-Norm | 88.95 | 94.47 | **70.79** | **83.17** | 58.34 | 74.10 | 78.11 | 87.43 | 29.67 | 47.48 | 77.92 | 88.63 | 87.28 | **92.02** |
| ConvAP | | BL | **89.69** | 95.14 | 76.83 | 85.08 | 63.33 | 77.65 | 76.22 | **85.14** | 33.41 | 51.46 | 81.88 | **92.26** | **86.17** | 91.06 |
| | | ALL | 90.23 | 95.29 | 75.56 | 85.40 | 65.50 | 79.48 | 80.54 | 88.38 | 34.72 | 52.03 | 80.56 | 89.95 | 86.45 | 91.33 |
| | | AH | 90.10 | 95.16 | 78.41 | 86.35 | 66.17 | 79.44 | 76.49 | 84.73 | 33.41 | 50.24 | 83.53 | 91.60 | 86.10 | 91.11 |
| | | ALL-Norm | 89.58 | 94.98 | **77.46** | **85.40** | **63.92** | **78.28** | **76.62** | 84.86 | **34.39** | **51.71** | 81.55 | **92.26** | 86.04 | **91.11** |
| | | AH-Norm | 89.48 | **95.35** | 75.56 | 84.76 | 62.61 | 76.45 | 74.32 | 82.84 | 33.41 | 50.24 | **83.53** | 91.60 | 86.10 | **91.11** |

However, among AH and ALL, AH works well in some cases and ALL in other cases. To balance fluctuations and leverage the combined capabilities of both variants, we will try the following function in our future work,

Table 11: Observations follow from Table 10.

| Method | Dim | ANU pairs | St Lucia | | SVOX | | SVOX Night | | SVOX Overcast | | SVOX Rain | | SVOX Snow | | SVOX Sun | |
|---|---|---|---|---|---|---|---|---|---|---|---|---|---|---|---|---|
| | | | R@1 | R@5 | R@1 | R@5 | R@1 | R@5 | R@1 | R@5 | R@1 | R@5 | R@1 | R@5 | R@1 | R@5 |
| BoQ | 8192 | BL | 60.66 | 86.48 | 97.99 | **98.99** | 58.57 | 74.97 | **96.56** | **98.28** | **91.89** | **97.01** | 96.21 | **98.85** | 87.24 | 95.43 |
| | | ALL | 60.59 | 86.41 | 97.69 | 98.79 | 60.27 | 75.94 | 96.22 | 97.94 | 89.22 | 96.37 | 94.37 | 97.93 | 83.02 | 91.45 |
| | | AH | 61.75 | 86.61 | 98.15 | 98.94 | 65.25 | 81.41 | 96.56 | 97.94 | 91.68 | 96.69 | 96.67 | 98.51 | 88.76 | 95.55 |
| | | ALL-Norm | 60.66 | 86.13 | 97.72 | 98.75 | 57.47 | 72.54 | 94.61 | 96.79 | 88.37 | 95.84 | 93.68 | 97.47 | 83.49 | 90.98 |
| | | AH-Norm | **61.41** | **86.48** | 98 | 98.96 | **65.01** | **78.25** | 95.87 | 97.71 | 91.36 | 96.58 | 95.86 | 98.51 | 85.60 | 94.85 |
| MixVPR | 4096 | BL | **61.41** | **86.20** | 96.62 | 98.28 | 44.59 | 63.79 | 93.23 | 97.13 | 86.45 | 93.38 | 92.53 | 97.24 | 77.87 | 86.77 |
| | | ALL | 61.07 | 86.41 | 96.81 | 98.40 | 52 | 70.84 | 93.69 | 97.59 | 88.58 | 94.77 | 93.56 | 98.05 | 79.63 | 89.58 |
| | | AH | 60.66 | 86.13 | 97.07 | 98.38 | 50.91 | 69.02 | 94.27 | 97.48 | 88.05 | 94.45 | 92.64 | 98.05 | 81.15 | 90.52 |
| | | ALL-Norm | 60.72 | 86.07 | 96.34 | 98.11 | **45.32** | 63.18 | 92.09 | 96.67 | 84.85 | 92.96 | 91.61 | 97.24 | 76.11 | 85.25 |
| | | AH-Norm | 61.20 | 85.79 | 96.29 | 97.97 | 41.80 | 58.20 | 91.74 | 96.79 | 83.56 | 92.53 | 89.54 | 96.32 | 76.46 | 85.71 |
| CosPlace | | BL | 59.90 | 85.66 | **96.57** | 98.28 | 32.44 | 50.67 | **92.09** | **97.02** | 80.26 | 90.50 | **90.34** | **97.36** | 69.67 | 83.26 |
| | | ALL | 59.77 | 85.93 | 96.79 | 98.42 | 40.22 | 59.78 | 93.81 | 97.25 | 84.53 | 92.10 | 92.41 | 97.24 | 74.24 | 85.95 |
| | | AH | 60.45 | 86.07 | 96.69 | 98.49 | 36.33 | 55.89 | 92.43 | 96.10 | 82.39 | 91.14 | 91.38 | 96.78 | 72.01 | 82.44 |
| | | ALL-Norm | 59.56 | 85.66 | 96.41 | 98.24 | **33.29** | **51.64** | 91.40 | 95.87 | **82.39** | **91.14** | 90.11 | 96.90 | 68.85 | 82.08 |
| | | AH-Norm | **60.04** | **85.72** | 96.55 | **98.34** | 32.32 | 49.82 | **92.09** | 96.22 | 81.43 | 89.22 | 90 | 95.86 | 66.98 | 80.68 |
| ConvAP | 4096 | BL | 58.95 | 85.25 | **94.99** | **97.09** | 19.32 | 33.90 | **81.88** | **91.63** | 69.26 | 80.68 | 77.13 | 88.51 | **58.31** | **74.71** |
| | | ALL | 60.66 | 86 | 95.16 | 97.35 | 32.44 | 48.60 | 88.42 | 94.50 | 77.27 | 86.02 | 86.55 | 94.71 | 67.56 | 80.44 |
| | | AH | 60.38 | 85.59 | 95.22 | 97.29 | 20.53 | 33.17 | 82.91 | 90.14 | 68.94 | 80.79 | 76.78 | 88.28 | 60.30 | 76.35 |
| | | ALL-Norm | 59.56 | **85.31** | 94.89 | 96.97 | **20.05** | **35.72** | 80.50 | 90.48 | 69.16 | **81.96** | **77.70** | **89.66** | 58.20 | 73.54 |
| | | AH-Norm | **59.70** | 84.77 | 94.83 | 96.95 | 11.79 | 22.84 | 74.77 | 86.12 | 58.06 | 73.96 | 65.29 | 81.03 | 47.66 | 63.58 |

$$\mathcal{L}_{anu-h} = \mathcal{L}_{original} + \frac{1}{|Q|}\sum_{q\in Q}\sum_{p\in P_q}\left(\lambda_h^+\mathcal{L}_{pk'} + \lambda_r^+\sum_{m\in P'_q\setminus\{k',p\}}\mathcal{L}_{pm} + \lambda_h^-\mathcal{L}_{pl'} + \lambda_r^-\sum_{y\in N_q\setminus l'}\mathcal{L}_{py}\right),\tag{24}$$

$$k' = \operatorname*{argmin}_{k\in P'_q\setminus p}\operatorname{sim}(p,k),\ \ l' = \operatorname*{argmax}_{l\in P'_q\setminus p}\operatorname{sim}(p,l),\ \#(\text{min pos sim \& max neg sim})$$

where all hardest terms and the rest of the additional terms can be weighted by $\lambda_h^+$, $\lambda_r^+$, $\lambda_h^-$, and $\lambda_r^-$ and their contribution can be controlled according to the model and training complexity.

Table 12: Recall performance average over three runs with different seeds. The trend in performance in main paper is almost followed by the average recalls here.

| Method | ANU pairs | Pittsburgh30k | | Tokyo 24/7 | | Nordland | | MSLS (Val) | | Amstertime | | SPEDTest | | Eynsham | |
|---|---|---|---|---|---|---|---|---|---|---|---|---|---|---|---|
| | | R@1 | R@5 | R@1 | R@5 | R@1 | R@5 | R@1 | R@5 | R@1 | R@5 | R@1 | R@5 | R@1 | R@5 |
| BoQ | BL-Avg | 91.25±0.08 | 95.32±0.14 | 81.06±1.28 | 89.52±1.46 | 72.44±1.81 | 84.38±1.21 | 85.72±0.69 | **91.67±0.28** | 37.62±0.68 | 57.45±0.24 | 82.54±0.99 | 90.50±1.82 | 89.05±0.25 | 92.97±0.20 |
| | ALL-Avg | 90.94±0.24 | 95.31±0.23 | 80.32±2.51 | 89.53±0.83 | 71.77±0.65 | 83.70±0.53 | 85.54±0.81 | 90.95±0.14 | 36.89±1.37 | 55.88±1.10 | 80.67±2.06 | 90.17±1.82 | 88.62±0.21 | 92.73±0.09 |
| | AH-Avg | **91.47±0.33** | **95.50±0.16** | **84.55±1.43** | **91.53±1.03** | **77.18±1.56** | **87.34±0.73** | **86.49±0.48** | 91.31±0.21 | **39.84±2.12** | **59.49±1.78** | **84.07±1.19** | **92.20±0.42** | **89.40±0.27** | **93.16±0.10** |
| | AE-Avg | 90.06±0.24 | 94.86±0.14 | 74.71±3.55 | 83.81±1.65 | 64.33±1.80 | 78.51±1.33 | 83.56±0.61 | 89.91±0.55 | 33.44±1.04 | 52.82±0.66 | 76.61±1.51 | 86.71±0.81 | 87.43±0.45 | 91.95±0.27 |
| MixVPR | BL-Avg | 90.62±0.25 | **95.29±0.19** | 77.99±1.20 | 88.47±0.66 | 72.91±1.96 | 84.72±1.38 | 83.02±0.55 | 89.64±0.68 | 35.42±0.76 | 54.66±0.91 | **84.07±0.67** | **92.81±0.81** | 88.12±0.14 | 92.20±0.08 |
| | ALL-Avg | 90.87±0.24 | 95.25±0.05 | 77.77±1.68 | 89.21±0.32 | 74.36±1.67 | 85.91±1.3 | **84.27±0.21** | **90.54±0.54** | **37.39±0.29** | **56.37±0.53** | 82.59±1.21 | 91.93±0.72 | 88.11±0.19 | 92.23±0.09 |
| | AH-Avg | **91.13±0.31** | 95.25±0.11 | **80.63±0.32** | **89.42±1.20** | **77.23±1.14** | **87.61±0.82** | 84.19±1.02 | 90.54±0.82 | 37.32±0.08 | 56.07±0.80 | 83.69±0.66 | 92.42±0.59 | **88.39±0.14** | **92.46±0.07** |
| | AE-Avg | 89.67±0.12 | 95.01±0.06 | 71.53±1.03 | 84.87±1.60 | 60.11±0.38 | 75.32±0.61 | 82.16±0.98 | 89.14±0.82 | 33.31±0.90 | 52.31±0.69 | 76.66±0.25 | 88.36±0.67 | 87.07±0.16 | 91.74±0.13 |
| CosPlace | BL-Avg | 88.85±0.24 | 94.41±0.17 | 68.67±1.47 | 79.89±0.97 | 59.66±0.82 | 75.13±0.93 | 80.36±0.52 | 88.50±1.27 | 29.56±0.57 | 47.45±0.62 | 79.52±0.41 | 89.90±0.82 | 87.38±0.05 | 92.02±0.01 |
| | ALL-Avg | **89.24±0.05** | **94.47±0.16** | 71.01±1.85 | **83.17±0.00** | 63.66±1.18 | 78.37±0.85 | 81.80±0.39 | 89.05±0.59 | **30.79±1.26** | **49.89±1.19** | 78.64±0.42 | 88.96±0.76 | 87.38±0.16 | 91.92±0.17 |
| | AH-Avg | 89.14±0.33 | 94.43±0.10 | **71.54±1.31** | 82.11±2.40 | 63.12±1.54 | 78.25±1.58 | 81.49±0.36 | **89.05±0.82** | 29.84±0.57 | 46.83±0.94 | **79.62±1.07** | **90.34±0.99** | **87.76±0.09** | **92.34±0.22** |
| | AE-Avg | 88.43±0.13 | 94.26±0.13 | 61.58±1.39 | 76.40±0.80 | 49.04±0.17 | 65.93±0.55 | 79.41±0.08 | 87.61±0.87 | 27.51±0.24 | 45.34±0.51 | 73.04±1.32 | 84.62±0.69 | 85.94±0.05 | 91.14±0.03 |
| ConvAP | BL-Avg | 90.00±0.29 | **95.24±0.08** | 76.30±2.42 | 85.40±0.32 | 64.38±1.48 | 78.38±1.35 | 78.15±1.67 | 86.26±0.97 | 33.22±0.20 | 50.79±0.59 | 81.93±0.09 | **92.53±0.35** | **86.37±0.26** | **91.36±0.29** |
| | ALL-Avg | **90.15±0.08** | 95.14±0.16 | 75.56±1.27 | 85.29±0.18 | 65.47±0.37 | 79.51±0.50 | **80.09±0.41** | **87.88±0.48** | **34.77±0.73** | 51.79±0.21 | 80.56±0.33 | 90.11±0.28 | 86.27±0.24 | 91.32±0.10 |
| | AH-Avg | **90.15±0.05** | 95.20±0.07 | **78.62±0.96** | **87.30±0.85** | **66.90±0.70** | **80.09±0.76** | 77.48±1.01 | 85.81±1.08 | 33.52±0.13 | 50.38±0.31 | **83.42±0.19** | 92.15±0.68 | 86.16±0.07 | 91.17±0.06 |
| | AE-Avg | 89.34±0.11 | 94.62±0.09 | 63.81±2.56 | 77.67±0.80 | 53.43±0.17 | 70.43±0.37 | 78.65±0.48 | 86.58±0.44 | 29.75±0.29 | 46.48±0.66 | 75.40±0.34 | 88.47±0.44 | 84.57±0.13 | 90.29±0.05 |

Table 13: Observations follow from Table 12.

| Method | ANU pairs | St Lucia | | SVOX | | SVOX Night | | SVOX Overcast | | SVOX Rain | | SVOX Snow | | SVOX Sun | |
|---|---|---|---|---|---|---|---|---|---|---|---|---|---|---|---|
| | | R@1 | R@5 | R@1 | R@5 | R@1 | R@5 | R@1 | R@5 | R@1 | R@5 | R@1 | R@5 | R@1 | R@5 |
| BoQ | BL-Avg | 60.95±0.27 | 86.48±0.07 | 97.92±0.07 | 98.95±0.05 | 61.04±4.72 | 77.60±3.76 | **96.37±0.24** | 98.05±0.20 | **92.14±0.52** | **97.23±0.22** | 95.75±0.53 | **98.66±0.17** | **87.98±0.64** | 94.81±0.54 |
| | ALL-Avg | 60.43±0.73 | 86.41±0.34 | 97.65±0.06 | 98.75±0.03 | 61.93±5.27 | 76.91±4.56 | 95.95±0.37 | **98.13±0.17** | 90.14±1.40 | 96.58±0.18 | 94.87±0.59 | 98.05±0.20 | 84.31±1.48 | 92.19±0.76 |
| | AH-Avg | **61.68±0.12** | **86.59±0.31** | **98.03±0.11** | **98.96±0.04** | **67.00±3.93** | **80.84±2.00** | 96.14±0.63 | 97.79±0.27 | 92.07±0.43 | 96.94±0.27 | **95.94±0.69** | 98.36±0.48 | **87.98±0.76** | **94.97±0.72** |
| | AE-Avg | 61.11±0.72 | 86.34±0.19 | 97.15±0.06 | 98.59±0.03 | 55.37±4.12 | 72.62±3.69 | 95.07±0.63 | 97.32±0.29 | 87.41±1.77 | 94.56±0.56 | 93.87±0.81 | 97.82±0.40 | 81.62±1.64 | 91.49±1.25 |
| MixVPR | BL-Avg | 61.07±0.66 | 86.20±0.14 | 96.55±0.06 | 98.19±0.08 | 46.54±2.49 | 64.56±0.68 | 93.20±0.40 | 96.90±0.30 | 85.66±1.19 | 93.63±0.62 | 91.61±0.92 | 97.36±0.12 | 78.18±0.29 | 87.94±1.24 |
| | ALL-Avg | 60.48±0.51 | 86.29±0.14 | 96.93±0.13 | **98.40±0.07** | 49.98±2.00 | **68.53±2.63** | 93.77±0.07 | 97.25±0.35 | **88.65±0.23** | **94.95±0.16** | **93.45±0.31** | 97.93±0.12 | 80.44±0.82 | 89.93±0.35 |
| | AH-Avg | 60.98±0.34 | **86.38±0.24** | **97.00±0.07** | 98.39±0.04 | **50.59±0.91** | 67.80±1.06 | **94.12±0.48** | 97.25±0.23 | 84.24±1.37 | 92.10±0.75 | 93.33±0.64 | **97.97±0.07** | **80.99±2.11** | **90.21±1.23** |
| | AE-Avg | **61.25±0.81** | 86.13±0.14 | 96.36±0.24 | 98.21±0.12 | 42.52±2.50 | 61.97±1.22 | 93.04±0.54 | 96.37±0.24 | 84.24±1.37 | 92.10±0.75 | 91.38±0.12 | 97.32±0.29 | 73.38±1.11 | 85.79±0.25 |
| CosPlace | BL-Avg | 60.02±1.20 | 85.72±0.07 | 96.54±0.14 | 98.33±0.06 | 36.90±4.10 | 56.22±4.92 | 91.86±0.40 | 96.56±0.41 | 81.57±1.60 | 91.00±0.76 | 90.27±0.24 | 96.59±0.67 | 69.94±0.91 | 83.41±0.27 |
| | ALL-Avg | 60.54±0.89 | **86.25±0.28** | **96.80±0.01** | **98.47±0.05** | 42.21±2.90 | 62.29±2.27 | **93.85±0.29** | **97.06±0.33** | **84.88±0.81** | **92.56±1.09** | **93.18±0.75** | **97.24±0.12** | **75.26±1.06** | **86.34±0.33** |
| | AH-Avg | 60.68±0.46 | 86.16±0.08 | 96.79±0.15 | 98.43±0.06 | 35.52±1.21 | 54.47±2.25 | 92.93±0.46 | 96.71±0.54 | 83.53±1.12 | 91.50±0.62 | 91.88±0.52 | 97.17±0.57 | 72.83±0.77 | 84.15±1.64 |
| | AE-Avg | **61.39±0.31** | 85.93±0.19 | 96.08±0.17 | 98.17±0.05 | 36.66±0.50 | 56.14±0.56 | 92.70±0.17 | 96.52±0.17 | 81.29±1.26 | 90.82±0.84 | 90.50±0.43 | 96.98±0.13 | 67.33±0.35 | 81.30±1.16 |
| ConvAP | BL-Avg | 59.86±0.79 | **85.75±0.43** | 95.29±0.26 | **97.35±0.23** | 24.18±4.36 | 40.99±6.39 | 85.63±3.27 | 92.89±1.14 | 73.53±3.77 | 85.31±4.00 | 82.53±4.87 | 91.42±2.52 | 64.95±5.70 | 80.60±5.06 |
| | ALL-Avg | 60.27±0.35 | **85.75±0.33** | 95.17±0.03 | 97.32±0.03 | **30.01±3.27** | **48.08±1.98** | **87.62±1.20** | **94.19±0.35** | **77.23±1.34** | **87.48±1.29** | **86.28±0.57** | **93.79±0.98** | **67.52±1.70** | **81.46±1.76** |
| | AH-Avg | 60.11±0.31 | 85.57±0.04 | **95.34±0.10** | **97.35±0.07** | 19.56±0.84 | 33.05±0.55 | 83.48±0.53 | 90.94±0.80 | 69.76±0.73 | 82.03±1.28 | 77.63±0.77 | 89.01±0.75 | 60.89±0.92 | 76.78±0.96 |
| | AE-Avg | **60.50±0.55** | 85.36±0.34 | 94.01±0.04 | 96.84±0.08 | 27.74±0.99 | 46.74±0.82 | 86.96±0.35 | 93.62±0.24 | 77.06±1.02 | 88.47±0.28 | 84.67±0.41 | 93.51±0.34 | 64.83±2.31 | 79.63±1.07 |

Table 14: Average R1/R5 performance across all test datasets and number of wins/losses for easier comparison. We see that the proposed variants AH and ALL wins in majority of the cases.

| Model | ANU-Type | Avg recall | | # win cases/total datasets | |
|---|---|---|---|---|---|
| | | R@1 | R@5 | R@1 | R@5 |
| BoQ | BL | 80.72 | 89.56 | 2/14 | 5/14 |
| | ALL | 79.70 | 88.59 | 0/14 | 0/14 |
| | AH | **82.64** | **90.81** | **13/14** | **9/14** |
| | AE | 76.17 | 86.63 | 0/14 | 0/14 |
| SALAD | BL | 87.63 | 94.80 | 4/14 | 2/14 |
| | ALL | 87.47 | 94.54 | 1/14 | 1/14 |
| | AH | **88.17** | **95.07** | **8/14** | **13/14** |
| | AE | 85.24 | 92.94 | 1/14 | 0/14 |
| MixVPR | BL | 78.51 | 87.98 | 1/14 | 1/14 |
| | ALL | 78.89 | 88.78 | 5/14 | 6/14 |
| | AH | **79.73** | **88.84** | **7/14** | **8/14** |
| | AE | 74.79 | 85.99 | 1/14 | 0/14 |
| R2F | BL | 54.24 | 67.52 | 0/14 | 0/14 |
| | ALL | **60.84** | **73.92** | **14/14** | **14/14** |
| | AH | 57.68 | 71.58 | 0/14 | 0/14 |
| | AE | 57.72 | 71.44 | 0/14 | 0/14 |
| NetVLAD | BL | 76.70 | **87.42** | 5/14 | **6/14** |
| | ALL | 76.75 | 86.82 | 3/14 | 3/14 |
| | AH | **77.11** | 86.48 | **6/14** | 5/14 |
| | AE | 71.52 | 82.90 | 0/14 | 1/14 |
| CosPlace | BL | 72.72 | 84.95 | 0/14 | 2/14 |
| | ALL | **74.82** | **85.30** | **8/14** | **7/14** |
| | AH | 73.46 | 84.96 | 6/14 | 5/14 |
| | AE | 70.94 | 82.92 | 1/14 | 0/14 |
| ConvAP | BL | 69.24 | 80.97 | 0/14 | 1/14 |
| | ALL | **72.83** | **82.82** | **10/14** | **11/14** |
| | AH | 70.66 | 82.73 | 4/14 | 1/14 |
| | AE | 68.36 | 81.97 | 0/14 | 1/14 |
| SupVLAD | BL | **84.34** | 92.35 | 4/14 | **6/14** |
| | ALL | 84.16 | **93.61** | **7/14** | 3/14 |
| | AH | 83.97 | 92.30 | 2/14 | 1/14 |
| | AE | 84.32 | 93.08 | 4/14 | 5/14 |

## A.9 Statistical Reliability

We repeat experiments with a few representative models for multiple seeds and report average recalls in Tables 12 and 13.

Further, for easier comparison of the proposed variants with baseline approaches, we present average recalls across all test datasets and number of wins/losses in Table 14.

## A.10 Test Results Across Multiple Input Dimensions

The main paper discusses the performance of the approaches on $224 \times 224$ dimensional images. In this section, we test the VPR algorithms trained on $224 \times 224$ images across varying input dimensions, including $112 \times 112$ and $320 \times 320$. These results are presented in Tables 15 and 16. We observe that as the input dimension increases, the performance of the proposed framework continues to improve.

## A.11 Details of the Datasets Used in this Work

We provide details of the datasets utilized in this work.

### A.11.1 Pittsburgh 30k (P30k)

The Pittsburgh 30k dataset Arandjelovic et al. (2016) is a subset of the large-scale Pittsburgh 250k dataset Torii et al. (2013). The dataset is constructed from Google Street View panoramas, which are divided into equally sized perspective views. The query and reference galleries are captured in different years and at different times of day. From each panorama (resolution 6656×3328), 24 perspective images of size 640×480 are generated by

Table 15: Performance comparison of VPR models on varying input dimensions. We observe that as the image resolution increases, the proposed approach continues to improve.

| Method | Dim | ANU pairs | Pittsburgh30k | | Tokyo 24/7 | | Nordland | | MSLS (Val) | | Amstertime | | SPEDTest | | Eynsham | |
|---|---|---|---|---|---|---|---|---|---|---|---|---|---|---|---|---|
| | | | R@1 | R@5 | R@1 | R@5 | R@1 | R@5 | R@1 | R@5 | R@1 | R@5 | R@1 | R@5 | R@1 | R@5 |
| BoQ | 8192 | BL-224 | 91.21 | 95.36 | 80.32 | 89.21 | 73.14 | 84.56 | 85.14 | 91.89 | 36.83 | 57.40 | 81.55 | 88.63 | 88.79 | 92.74 |
| | | ALL-224 | 90.67 | 95.06 | 78.41 | 88.89 | 72.24 | 83.85 | 84.73 | 91.08 | 35.37 | 55.12 | 78.58 | 88.80 | 88.48 | 92.63 |
| | | AH-224 | 91.51 | 95.67 | 84.44 | 90.79 | 78.88 | 88.19 | 86.62 | 91.49 | 38.37 | 58.46 | 82.70 | 91.76 | 89.56 | 93.27 |
| | | AE-224 | 90.11 | 94.87 | 73.33 | 84.76 | 63.05 | 77.30 | 82.97 | 89.59 | 34.63 | 53.58 | 76.28 | 86.16 | 87.24 | 91.82 |
| | | BL-112 | 88.04 | 93.68 | 59.05 | 73.33 | 52.65 | 69.28 | 69.05 | 80.27 | 23.33 | 39.84 | 75.29 | 87.15 | 80.53 | 87.74 |
| | | ALL-112 | 87.50 | 93.47 | 55.87 | 73.02 | 48.75 | 65.22 | 68.78 | 79.73 | 21.79 | 38.54 | 70.35 | 86.82 | 79.18 | 86.93 |
| | | AH-112 | 88.34 | 93.93 | 66.67 | 78.73 | 57.59 | 72.83 | 73.24 | 82.43 | 25.77 | 42.11 | 75.95 | 88.47 | 82.21 | 88.98 |
| | | AE-112 | 86.28 | 92.74 | 53.02 | 66.98 | 41.64 | 59.96 | 65.95 | 77.70 | 19.67 | 36.26 | 67.38 | 80.72 | 75.94 | 84.75 |
| | | BL-320 | 91.15 | 95.72 | 85.71 | 91.43 | 75.01 | 85.56 | 87.70 | 92.97 | 40.16 | 60.33 | 79.74 | 87.64 | 90.48 | 94.01 |
| | | ALL-320 | 91.11 | 95.48 | 82.54 | 92.38 | 73.16 | 83.80 | 87.03 | 92.03 | 36.75 | 58.21 | 78.58 | 88.14 | 90.40 | 93.96 |
| | | AH-320 | 91.17 | 95.48 | 87.30 | 93.02 | 80.62 | 88.90 | 88.11 | 93.11 | 42.11 | 62.28 | 81.38 | 91.27 | 90.80 | 94.25 |
| | | AE-320 | 90.14 | 94.91 | 76.51 | 87.30 | 62.23 | 75.80 | 85.81 | 91.89 | 35.37 | 55.20 | 75.95 | 86.49 | 89.29 | 93.23 |
| SALAD | 4096 | BL-224 | 92.06 | 96.16 | 91.75 | 97.14 | 78.88 | 89.78 | 91.22 | 95.41 | 52.28 | 74.15 | 90.44 | 95.88 | 90.53 | 94.48 |
| | | ALL-224 | 91.86 | 96.16 | 93.33 | 96.83 | 78.66 | 88.88 | 90.0 | 95.54 | 53.01 | 73.66 | 90.28 | 94.56 | 90.55 | 94.63 |
| | | AH-224 | 92.63 | 96.39 | 93.65 | 97.46 | 79.21 | 89.96 | 90.68 | 95.68 | 55.28 | 75.77 | 89.62 | 95.22 | 91.03 | 94.73 |
| | | AE-224 | 90.68 | 95.32 | 87.62 | 94.92 | 67.55 | 80.18 | 89.86 | 94.73 | 47.24 | 69.11 | 87.64 | 93.90 | 90.06 | 94.30 |
| | | BL-112 | 85.14 | 93.66 | 63.49 | 79.68 | 26.86 | 43.95 | 72.16 | 86.22 | 24.96 | 43.74 | 62.11 | 77.43 | 74.61 | 58.10 |
| | | ALL-112 | 85.23 | 93.35 | 64.13 | 78.41 | 28.55 | 44.19 | 72.30 | 85 | 26.34 | 44.80 | 66.89 | 82.54 | 74.28 | 85.01 |
| | | AH-112 | 86.61 | 93.79 | 64.76 | 78.73 | 26.08 | 41.88 | 72.97 | 85.81 | 27.15 | 46.34 | 66.23 | 80.56 | 76.06 | 85.86 |
| | | AE-112 | 83.69 | 92.71 | 58.10 | 75.87 | 18.14 | 30.57 | 71.49 | 83.65 | 22.22 | 38.78 | 54.37 | 70.51 | 71.26 | 82.93 |
| | | BL-448 | 92.09 | 96.24 | 94.60 | 97.78 | 88.09 | 94.72 | 92.16 | 95.95 | 59.35 | 80.89 | 91.27 | 95.22 | 91.51 | 95.04 |
| | | ALL-448 | 91.87 | 96.23 | 95.56 | 98.10 | 88.89 | 95.32 | 91.89 | 96.35 | 59.43 | 79.84 | 89.79 | 94.73 | 91.69 | 95.07 |
| | | AH-448 | 92.52 | 96.58 | 96.51 | 98.10 | 88.83 | 95.31 | 93.24 | 96.62 | 63.01 | 82.76 | 90.44 | 95.22 | 92.05 | 95.42 |
| | | AE-448 | 90.57 | 95.75 | 91.11 | 96.83 | 77.28 | 86.87 | 90.95 | 95.95 | 54.88 | 76.91 | 89.29 | 94.89 | 91.32 | 94.89 |
| CosPlace | 1024 | BL-224 | 89.10 | 94.51 | 66.98 | 79.05 | 60.25 | 75.59 | 80 | 89.97 | 29.51 | 47.64 | 79.57 | 89.79 | 87.42 | 92.02 |
| | | ALL-224 | 89.22 | 94.60 | 68.89 | 83.17 | 64.36 | 78.97 | 82.03 | 89.46 | 29.35 | 48.78 | 78.58 | 88.80 | 87.28 | 91.73 |
| | | AH-224 | 89.51 | 94.41 | 71.11 | 84.44 | 63.53 | 78.74 | 81.62 | 89.46 | 30.49 | 47.80 | 80.72 | 91.43 | 87.75 | 92.22 |
| | | AE-224 | 88.54 | 94.18 | 60.63 | 75.56 | 48.85 | 65.78 | 79.46 | 88.24 | 27.56 | 45.04 | 74.46 | 84.84 | 85.90 | 91.11 |
| | | BL-112 | 84.14 | 92.40 | 47.62 | 65.71 | 34.39 | 51.81 | 61.49 | 73.24 | 20.16 | 32.76 | 68.86 | 83.69 | 77.53 | 86.28 |
| | | ALL-112 | 84.86 | 92.15 | 50.16 | 64.76 | 35.89 | 54.01 | 64.05 | 75.14 | 20.73 | 35.28 | 67.05 | 80.23 | 76.65 | 85.62 |
| | | AH-112 | 84.84 | 92.40 | 47.62 | 66.67 | 39.37 | 57.64 | 64.32 | 77.03 | 19.27 | 36.26 | 68.53 | 82.70 | 78.04 | 86.85 |
| | | AE-112 | 82.89 | 91.36 | 39.05 | 61.59 | 23.53 | 39.08 | 58.24 | 70.81 | 16.75 | 31.71 | 59.64 | 76.44 | 72.02 | 82.61 |
| | | BL-320 | 89.17 | 94.60 | 72.70 | 82.54 | 61.99 | 76.60 | 82.97 | 89.05 | 32.85 | 50.24 | 78.75 | 88.47 | 88.83 | 93.14 |
| | | ALL-320 | 90.23 | 95.20 | 75.24 | 82.86 | 66.41 | 79.68 | 84.73 | 90.81 | 33.09 | 52.93 | 79.08 | 89.13 | 88.94 | 93.06 |
| | | AH-320 | 89.77 | 94.76 | 76.83 | 85.71 | 62.69 | 77.43 | 83.92 | 90.54 | 32.03 | 49.84 | 80.89 | 91.60 | 89.27 | 93.39 |
| | | AE-320 | 89.25 | 94.50 | 66.98 | 79.37 | 53.34 | 69.21 | 83.51 | 89.59 | 30 | 48.05 | 74.96 | 85.34 | 88 | 92.48 |
| ConvAP | 4096 | BL-224 | 89.69 | 95.14 | 76.83 | 85.08 | 63.33 | 77.65 | 76.22 | 85.14 | 33.41 | 51.46 | 81.88 | 92.26 | 86.17 | 91.06 |
| | | ALL-224 | 90.23 | 95.29 | 75.56 | 85.40 | 65.50 | 79.48 | 80.54 | 88.38 | 34.72 | 52.03 | 80.56 | 89.95 | 86.45 | 91.33 |
| | | AH-224 | 90.10 | 95.16 | 78.41 | 86.35 | 66.17 | 79.44 | 76.49 | 84.73 | 33.41 | 50.24 | 83.53 | 91.60 | 86.10 | 91.11 |
| | | AE-224 | 89.41 | 94.72 | 63.17 | 76.83 | 53.53 | 70.84 | 78.24 | 86.08 | 30.08 | 47.24 | 75.78 | 88.63 | 84.72 | 90.35 |
| | | BL-112 | 85.01 | 92.78 | 54.29 | 71.75 | 37.03 | 53.09 | 62.03 | 71.89 | 23.98 | 38.94 | 73.64 | 87.97 | 78.02 | 86.40 |
| | | ALL-112 | 85.52 | 93.05 | 50.16 | 68.89 | 36.89 | 53.58 | 61.89 | 73.78 | 22.36 | 36.18 | 71.17 | 84.51 | 77.49 | 86.26 |
| | | AH-112 | 85.80 | 93.12 | 55.56 | 73.02 | 41.60 | 58.25 | 64.05 | 74.32 | 24.88 | 41.46 | 75.12 | 88.30 | 79.30 | 87.25 |
| | | AE-112 | 83.55 | 92.27 | 37.14 | 55.56 | 24.53 | 41.26 | 57.43 | 70.68 | 16.34 | 30.08 | 65.24 | 80.56 | 70.67 | 82.26 |
| | | BL-320 | 89.47 | 95.50 | 77.78 | 88.25 | 65.16 | 79.34 | 76.62 | 83.78 | 36.18 | 55.53 | 80.56 | 90.94 | 87.16 | 91.77 |
| | | ALL-320 | 90.40 | 95.50 | 79.05 | 86.35 | 68.28 | 81.14 | 80.14 | 87.43 | 36.26 | 56.34 | 79.90 | 90.44 | 87.98 | 92.16 |
| | | AH-320 | 89.98 | 95.50 | 79.68 | 88.25 | 68.16 | 80.74 | 78.24 | 84.05 | 34.96 | 52.52 | 83.53 | 92.09 | 86.99 | 91.56 |
| | | AE-320 | 89.35 | 94.94 | 67.30 | 78.73 | 54.45 | 70.96 | 78.51 | 87.03 | 32.36 | 50.08 | 76.11 | 87.48 | 86.35 | 91.49 |

varying the yaw and pitch angles. The Pittsburgh 250k dataset primarily introduces viewpoint variations between query and reference images, with negligible illumination or seasonal changes.

## A.11.2 Nordland

The Nordland dataset is derived from a documentary recording of a 729 km railway journey, specifically covering the 182 km stretch between Trondheim and Bod in Norway, captured across all four seasons. A camera mounted on the front of the train recorded the same scenes across seasons: summer, winter, spring, and fall. The video frames were processed to remove segments containing tunnels and stations, and the remaining frames were geo-tagged. The Nordland dataset was specifically introduced for studying appearance-invariant feature learning and also presents challenges of visual aliasing due to numerous visually similar scenes along the route.

Table 16: Observations follow from Table 15.

| Method | Dim | ANU pairs | St Lucia | | SVOX | | SVOX Night | | SVOX Overcast | | SVOX Rain | | SVOX Snow | | SVOX Sun | |
|---|---|---|---|---|---|---|---|---|---|---|---|---|---|---|---|---|
| | | | R@1 | R@5 | R@1 | R@5 | R@1 | R@5 | R@1 | R@5 | R@1 | R@5 | R@1 | R@5 | R@1 | R@5 |
| BoQ | 8192 | BL-224 | 60.66 | 86.48 | 97.99 | 98.99 | 58.57 | 74.97 | 96.56 | 98.28 | 91.89 | 97.01 | 96.21 | 98.85 | 87.24 | 95.43 |
| | | ALL-224 | 60.59 | 86.41 | 97.69 | 98.79 | 60.27 | 75.94 | 96.22 | 97.94 | 89.22 | 96.37 | 94.37 | 97.93 | 83.02 | 91.45 |
| | | AH-224 | 61.75 | 86.61 | 98.15 | 98.94 | 65.25 | 81.41 | 96.56 | 97.94 | 91.68 | 96.69 | 96.67 | 98.51 | 88.76 | 95.55 |
| | | AE-224 | 61 | 86.41 | 97.11 | 98.57 | 50.67 | 68.41 | 94.95 | 97.36 | 85.38 | 93.92 | 93.79 | 98.05 | 79.86 | 90.05 |
| | | BL-112 | 58.95 | 84.56 | 93.81 | 97 | 17.86 | 32.56 | 85.89 | 93.58 | 72.25 | 84.10 | 78.74 | 89.31 | 57.73 | 71.43 |
| | | ALL-112 | 59.36 | 83.95 | 92.89 | 96.50 | 17.01 | 31.47 | 86.01 | 93.69 | 72.47 | 83.56 | 78.74 | 89.08 | 54.33 | 68.27 |
| | | AH-112 | 60.18 | 84.77 | 94.12 | 97.03 | 18.83 | 32.81 | 87.39 | 93.35 | 74.60 | 84.74 | 81.72 | 91.38 | 59.84 | 72.48 |
| | | AE-112 | 59.02 | 83.40 | 90.87 | 95.41 | 14.46 | 24.91 | 81.31 | 90.94 | 66.60 | 81.32 | 74.60 | 87.13 | 50.47 | 66.86 |
| | | BL-320 | 60.38 | 86.54 | 98.40 | 99.19 | 68.89 | 82.38 | 97.36 | 99.08 | 93.17 | 97.97 | 97.70 | 99.31 | 91.10 | 96.49 |
| | | ALL-320 | 60.93 | 86.20 | 98.18 | 99.01 | 69.14 | 83.23 | 97.13 | 98.74 | 90.29 | 96.69 | 96.32 | 98.85 | 88.64 | 95.43 |
| | | AH-320 | 60.66 | 86.68 | 98.42 | 99.18 | 71.20 | 85.78 | 97.59 | 98.85 | 93.60 | 98.29 | 97.47 | 99.66 | 93.68 | 98.13 |
| | | AE-320 | 59.63 | 86.34 | 97.63 | 98.82 | 60.63 | 77.64 | 94.37 | 97.93 | 94.61 | 98.17 | 87.41 | 93.92 | 83.26 | 92.39 |
| SALAD | 4096 | BL-224 | 60.04 | 86.75 | 98.09 | 99.15 | 93.32 | 98.06 | 97.94 | 99.08 | 97.55 | 99.36 | 98.28 | 99.66 | 94.50 | 98.13 |
| | | ALL-224 | 59.08 | 86.68 | 97.93 | 99.12 | 90.52 | 97.21 | 97.13 | 99.08 | 97.65 | 99.47 | 98.16 | 99.54 | 94.38 | 98.13 |
| | | AH-224 | 59.29 | 86.95 | 98.24 | 99.26 | 94.29 | 98.66 | 97.71 | 99.31 | 97.44 | 99.47 | 98.85 | 99.66 | 94.50 | 98.48 |
| | | AE-224 | 59.63 | 86.41 | 97.76 | 99.03 | 90.04 | 97.57 | 97.59 | 98.85 | 97.33 | 99.04 | 97.59 | 99.43 | 94.73 | 98.36 |
| | | BL-112 | 55.87 | 83.13 | 93.65 | 97.54 | 27.58 | 45.93 | 83.72 | 91.86 | 71.18 | 85.27 | 71.49 | 85.63 | 60.77 | 77.87 |
| | | ALL-112 | 56.56 | 82.86 | 92.78 | 97.12 | 29.65 | 43.62 | 81.31 | 92.20 | 70.76 | 82.28 | 70.80 | 86.55 | 62.76 | 77.75 |
| | | AH-112 | 56.01 | 83.47 | 93.69 | 97.51 | 27.46 | 44.23 | 82.80 | 93.46 | 71.50 | 83.24 | 70.69 | 86.21 | 61.71 | 77.52 |
| | | AE-112 | 56.22 | 82.86 | 92.53 | 96.94 | 24.79 | 43.62 | 80.62 | 91.17 | 68.94 | 83.78 | 69.43 | 83.91 | 59.84 | 77.05 |
| | | BL-448 | 59.90 | 86.75 | 98.36 | 99.26 | 96.84 | 99.15 | 98.17 | 99.20 | 98.08 | 99.57 | 99.20 | 99.66 | 96.96 | 99.30 |
| | | ALL-448 | 59.36 | 86.54 | 98.17 | 99.26 | 96.72 | 98.66 | 97.59 | 98.97 | 97.12 | 99.57 | 98.85 | 99.54 | 96.37 | 99.30 |
| | | AH-448 | 59.56 | 86.89 | 98.51 | 99.38 | 97.33 | 99.51 | 98.28 | 99.43 | 98.19 | 99.47 | 99.08 | 99.66 | 97.54 | 99.30 |
| | | AE-448 | 60.18 | 86.89 | 98.14 | 99.12 | 95.50 | 98.54 | 97.82 | 99.08 | 95.94 | 99.25 | 98.85 | 99.66 | 96.25 | 99.30 |
| CosPlace | 1024 | BL-224 | 59.90 | 85.66 | 96.57 | 98.28 | 32.44 | 50.67 | 92.09 | 97.02 | 80.26 | 90.50 | 90.34 | 97.36 | 69.67 | 83.26 |
| | | ALL-224 | 59.77 | 85.93 | 96.79 | 98.42 | 40.22 | 59.78 | 93.81 | 97.25 | 84.53 | 92.10 | 92.41 | 97.24 | 74.24 | 85.95 |
| | | AH-224 | 60.45 | 86.07 | 96.69 | 98.49 | 36.33 | 55.89 | 92.43 | 96.10 | 82.39 | 91.14 | 91.38 | 96.78 | 72.01 | 82.44 |
| | | AE-224 | 61.68 | 85.72 | 96.14 | 98.17 | 36.82 | 55.53 | 92.66 | 96.33 | 82.07 | 91.36 | 90 | 96.90 | 67.33 | 82.08 |
| | | BL-112 | 58.61 | 81.69 | 88.04 | 93.73 | 11.91 | 22.36 | 71.67 | 84.63 | 52.93 | 69.58 | 61.49 | 79.66 | 34.89 | 51.52 |
| | | ALL-112 | 58.74 | 82.04 | 88.85 | 94.21 | 12.88 | 25.03 | 75.23 | 88.30 | 58.80 | 74.07 | 67.93 | 83.22 | 35.36 | 50 |
| | | AH-112 | 59.08 | 82.58 | 89.35 | 94.66 | 12.27 | 23.57 | 76.03 | 87.50 | 58.06 | 72.15 | 68.39 | 82.30 | 37.24 | 52.11 |
| | | AE-112 | 56.35 | 80.05 | 85.33 | 92.16 | 10.45 | 22.24 | 70.30 | 86.35 | 52.29 | 70.54 | 62.07 | 80.34 | 30.44 | 45.43 |
| | | BL-320 | 59.22 | 85.79 | 97.24 | 98.67 | 35.84 | 55.16 | 94.04 | 97.1 3 | 86.23 | 94.13 | 92.76 | 97.82 | 77.05 | 89.34 |
| | | ALL-320 | 59.08 | 86.61 | 97.53 | 98.84 | 49.45 | 68.29 | 94.15 | 97.59 | 88.15 | 94.56 | 94.60 | 97.93 | 82.90 | 91.69 |
| | | AH-320 | 59.56 | 86.48 | 97.51 | 98.82 | 44.47 | 62.45 | 93 | 96.90 | 87. 62 | 93.38 | 92.53 | 96.90 | 78.34 | 87.35 |
| | | AE-320 | 58.20 | 86 | 97.10 | 98.64 | 40.95 | 62.21 | 92.66 | 96.90 | 85.59 | 93.38 | 92.87 | 97.59 | 78.45 | 88.64 |
| ConvAP | 4096 | BL-224 | 58.95 | 85.25 | 94.99 | 97.09 | 19.32 | 33.90 | 81.88 | 91.63 | 69.26 | 80.68 | 77.13 | 88.51 | 58.31 | 74.71 |
| | | ALL-224 | 60.66 | 86 | 95.16 | 97.35 | 32.44 | 48.60 | 88.42 | 94.50 | 77.27 | 86.02 | 86.55 | 94.71 | 67.56 | 80.44 |
| | | AH-224 | 60.38 | 85.59 | 95.22 | 97.29 | 20.53 | 33.17 | 82.91 | 90.14 | 68.94 | 80.79 | 76.78 | 88.28 | 60.30 | 76.35 |
| | | AE-224 | 59.90 | 85.04 | 93.98 | 96.75 | 26.85 | 46.05 | 87.27 | 93.35 | 76.63 | 88.79 | 85.06 | 94.25 | 62.41 | 78.69 |
| | | BL-112 | 57.86 | 80.94 | 87.08 | 92.55 | 9.60 | 19.68 | 71.67 | 84.75 | 51.87 | 67.34 | 60.34 | 79.20 | 35.48 | 51.29 |
| | | ALL-112 | 58.20 | 81.97 | 86.76 | 92.71 | 12.39 | 20.66 | 72.25 | 84.63 | 54.54 | 72.15 | 61.03 | 79.54 | 37.94 | 53.16 |
| | | AH-112 | 59.22 | 82.10 | 88.02 | 93.46 | 10.57 | 20.29 | 72.82 | 85.78 | 53.58 | 69.90 | 61.95 | 80.92 | 40.05 | 53.63 |
| | | AE-112 | 56.63 | 80.19 | 83.38 | 90.82 | 9.72 | 20.05 | 66.17 | 81.65 | 49.41 | 67.02 | 56.09 | 75.06 | 34.66 | 51.17 |
| | | BL-320 | 58.61 | 85.79 | 95.45 | 97.41 | 14.95 | 28.68 | 76.83 | 87.27 | 65.21 | 77.59 | 70 | 82.18 | 55.74 | 72.25 |
| | | ALL-320 | 59.49 | 86.41 | 95.87 | 97.59 | 23.45 | 43.13 | 83.72 | 91.40 | 70.54 | 83.03 | 80.23 | 89.77 | 64.87 | 79.98 |
| | | AH-320 | 59.49 | 85.72 | 95.94 | 97.71 | 17.13 | 31.23 | 78.44 | 87.73 | 64.35 | 80.15 | 71.03 | 83.68 | 57.85 | 72.01 |
| | | AE-320 | 59.29 | 85.72 | 94.90 | 97.09 | 16.65 | 33.05 | 78.56 | 90.83 | 71.50 | 85.17 | 75.40 | 90 | 57.26 | 74.71 |

### A.11.3 Tokyo 24/7

Tokyo 24/7 offers significant illumination variation between query and reference images due to day-night shifts. This is comprised of 76k reference images and 315 query images. The reference images are acquired from Google Street View during daytime, and the query images are captured by a mobile phone camera at different times of the day including daylight, sunset, and night. The Tokyo 24/7 dataset is entirely used for testing without train and val splits.

### A.11.4 MSLS

Most existing datasets lack broad geographical coverage and visual diversity, and some datasets do not contain accurate viewpoint information. To cater to this, the MSLS dataset proposes a dataset with a wide range of variations, including weather changes, day-night shifts, appearance changes due to seasonal variations, structural changes resulting from temporal variations, and distracting transients (such as cars and pedestrians). This is achieved by spanning the data collection across six continents, covering nearly 60 cities, over a period of seven years. Furthermore, this approach utilises different camera sensors to capture sensor variations. Importantly, MSLS contains sequences for sequence-based feature learning. In addition,

MSLS work proposes different image retrieval techniques, including im2im, im2seq, seq2im, and seq2seq. Image-to-image (Img2im) and sequence-to-sequence (Seq2Seq) approaches are common in the visual place recognition literature. In the im2im case, query and reference images are both non-sequential, standalone images. Seq2seq is where the top1 retrieved sequence is averaged using various other unsupervised techniques. Seq2im follows a majority voting approach, where each query has its top-1, the database image that is the closest to many frames in the query sequence; then that is the top-1. In the im2seq case, the sequence with the closest reference image to the query image becomes the top1 reference sequence.

### A.11.5 GSV-Cities

Most datasets for image-level descriptor-based VPR models contain noisy geographical labels, as well as other drawbacks, including limited geographical coverage and temporal variations. To address this, the GSV-Cities dataset was introduced, which includes diverse and challenging scenarios that occur over time, such as seasons, viewpoints, illumination, and occlusion. The dataset spans 40 cities across all continents over 14 years, providing accurate ground truths. Most model approaches trained on the GSV-Cities dataset generalise well on many test datasets. The dataset is entirely used for training without any validation or test splits.

### A.11.6 Oxford

Oxford dataset is collected by traversing through the central Oxford area, UK, twice every week for a period of a year from 2014 to 2015. The data is collected from six different cameras and various other sensors, such as GPU, INS, etc., mounted on a RobotCar from around 100 traversals for a total distance of 1000 km. The dataset includes a wide range of weather, seasonal, day-night illumination changes, construction variation, etc., over a long period traversal through the same route at different times and conditions. It contains video sequences.

### A.11.7 CUBS-200-2011

The CUBS-200-2011 dataset Wah et al. (2011) is an extension of the CUB-200 dataset, containing 11,788 images across 200 bird species, collected from Flickr. Ground truth labels, fine-grained labels of the birds, and bounding-box co-ordinates are obtained by Amazon Mechanical Turk (MTurk). This dataset is specifically introduced for applications such as fine-grained classification, which is also called subordinate categorization in cognitive science, multi-class object detection or part-based methods, image retrieval, etc. Fine-grained classification is a granular level classification of objects, where within a class, we identify a subclass. For example, identifying the specific category of the broad bird class, such as pelicans vs. sparrows. Part-based methods first detect and locate different parts of an object and later classify the whole image based on the parts.

### A.11.8 SOP

The Stanford Online Products dataset contains 120k images of 23k classes. The images are web crawled from eBay.com and deduplicated. This was primarily proposed for deep metric learning.

### A.11.9 Cars-196

This dataset was introduced for the task of fine-grained categorization. It comprises 16,185 images representing 196 different car categories. The images are sourced from the web, specifically from Flickr, Bing, and Google. They are uploaded to MTurk to determine their models.

### A.11.10 Amstertime

The Amstertime dataset Yildiz et al. (2022) is a collection of historical archival images from the city of Amsterdam, along with corresponding street-view images. The images contain viewpoint, scale, color, and camera lens variations, and occlusion. The authors have developed a custom crowd sourcing website that displays a combination of collected historical images and the 3D scene pointing a navigator from the Mapillary

platform towards the place from where the archival image was captured from. The user is expected to find matching images from these sets. Further verification is done to filter false matching pairs. The archival images are selected from the well-documented Beeldbank repository of the Amsterdam City Archives [2].

### A.11.11   SPED Test

The Specific PlacEs Dataset (SPED) dataset was originally proposed by Chen et al. (2017) to enable large scale VPR training. This dataset is subset of images collected from various publicly accessible outdoor surveillance cameras. The subset is curated by filtering dark images and structured to cover varying environmental conditions and landscapes. The SPED dataset randomly selected the 2543 cameras from around 30K cameras. The SPED Test Chen et al. (2018) dataset is constructed from 668 manually selected cameras from those used for the original large scale dataset. It simply acts as a test split.

### A.11.12   Eynsham

The Eynsham dataset Cummins & Newman (2009) was collected by traversing the same route multiple times covering 70km overall, 35km each time. This is by using a car-mounted camera. Images from the first traversal are used as the reference database and from the second traversal as queries.

### A.11.13   SVOX

The Street View OXford dataset Berton et al. (2021) is a large scale dataset depicting the city of Oxford is constructed using the Google Street View imagery. This primarily constitutes the reference database. The query image sets of multiple domains are taken from Oxford RobotCar dataset Maddern et al. (2017) and compared against the SVOX reference database. The query domains include, Snow, Rain, Sun, Night, and Overcast. This dataset is primarily to address the problem of domain shift in VPR.

---

[2]https://archief.amsterdam/beeldbank/

