# OpenReview forum: "ANU-RL: A New Perspective on Unsupervised Representation Learning for Visual Place Recognition"
_TMLR — Under review for TMLR_

### Review · Reviewer_ZQV9 · 2026-05-05

**Summary Of Contributions:**

This paper presents a framework for unsupervised representation learning for visual place recognition, which aims to embed the data in a new space to minimize intra-class distance and maximize inter-class distance. The framework explicitly includes relationships between positives and negatives, and within positives. Within the framework, the paper also considers various variants such as ANU-easy, ANU-hard as well as various contrastive loss functions such as MSim loss, Triplet loss and FastAP loss. The paper gives details on the implementations and evaluations of the proposed framework. Extensive experimental results are also given to illustrate the practical behavior of the proposed methods.

**Strength**
- Most existing methods focus only on the relationship between query-positives and query-negatives. The proposed framework considers various additional relationships to further improve the efficiency of representation learning in visual place recognition.
- The proposed framework is flexible as it immediately gives several variants. Furthermore, it allows the incorporation of various loss functions.
- The paper gives comprehensive experimental analyses, such as quantitative analyses, qualitative analyses. Comparisons with various baseline methods on various datasets are also included to justify the efficiency of the proposed method.

**Weakness**
- The description of the framework is a bit abstract and not clear. For example, in equation (1), there is $\sum_k L_{qk}+\sum_l L_{ql}$. Then, there is no difference between the positive and negative examples as the objective treats positive and negative examples in a similar way. From this objective function, it is not clear to me how it aims to minimize intra-class distance and maximize inter-class distance.
- In Eq (3), the objective function involves $L_{pk'}+L_{pl'}$. Then, the argmin similarity and argmax similarity is treated in the same way. It is not clear to me how this relates to min positive similarity and max negative similarity. Finally, I cannot see the difference between Eq (3) and Eq (4). Due to the symmetry, it seems that these two objective functions are essentially the same.
- In the abstract, the authors mention that the proposed method comes at no extra cost at inference time. I am wondering if the consideration of other relationships bring additional computational costs in solving the optimization problem.

**Audience:**

Yes

**Audience Explanation:**

Representation learning is an effective method to embed the data to relevant space for efficiency. The proposed framework is flexible and efficient, which should be interesting to the TMLR community.

**Broader Impact Concerns:**

I do not have broader impact concerns.

**Claims And Evidence:**

Yes

**Claims Explanation:**

The proposed method is backed up with extensive empirical analyses. The paper also includes details on the implementation and evaluations Meanwhile, there are several unclear descriptions of the proposed framework in equations. See points in weakness.

**Requested Changes:**

The authors should clarify the descriptions of the framework in equations. In particular, it is beneficial to clarify the difference in considering the positive and negative examples in the formulation.

Eq (11): it is not clear to me the meaning of $H_j^+$ and $h_j^+$.

Eq (10) uses a different meaning of $P_q'$ as compared to Eq (2). The meaning of notations should be consistent.

Introduction: "it's positive" should be "its positive". Similar change should be also made to "it's negative"

Eq (9) uses $|d_{qp+m-d_{qn}}|$. First, the meaning of $q_{qp}$ is not given. Second, there is an absolute value. Then, it is not necessary to use $()_+$, which according to the traditional notation, means the positive part.

---

> ### Author Response · Authors · 2026-07-05
> **Our response clarifies inconsistencies in some of the notations and explanations for terminology used in our work**
>
> We thank you for your positive feedback, for pointing out several inconsistencies in the notations, and for seeking clarity on certain terminology. Our detailed response to these comments is as follows.
>
> **Weakness 1:** Section 3. of the main paper has been revised for better clarity. The framework in Eq. (1) serves as an abstraction of the general contrastive loss setting. This alone does not constitute the final expression in which we directly optimize distance scores. Rather, the expressions of $L_{qk}$ and $L_{ql}$ differ within the framework depending on the specific objective, exhibiting a contrastive nature. Some of these involve exponential terms, and others are distance metrics. For instance, Multi-Sim loss contains exponential expressions in which the positive and negative similarities in the exponents have opposite signs, indicating contrastive behavior. We used this expression primarily to distinguish the template that existing loss functions follow from the proposed one.
>
> **Weakness 2:** Thank you for pointing this out. The correct expression should be $l\in{N}_q$ (and not $l\in{P^\prime}_q\backslash p$). We hope this clarifies it now.
>
> **Weakness 3:** This is true: inference incurs no additional costs, whereas training requires additional computations due to additional relationships, increasing training time. Our implementation uses loop-based scripts, which is not optimal. However, advanced dynamic programming concepts can parallelize and speed up the computations. As mentioned, once training finishes, the objective function plays no role in inference. Hence, no change in the inference time.
>
> **Requested Changes 1**: Please refer to the response to **Weakness 1**
>
> **Requested Changes 2**:  Meaning of $H_j^+$ and $h_j^+$.
>
> **Response**: This has been included in Appendix A.4.1 of the revised manuscript. In general, Average Precision (AP) is used as an evaluation metric that we typically aim to maximize. However, instead of limiting it to evaluation, recent research has tried to use AP directly as the training objective. The fastAP loss is one such approach. However, AP in its original form is non-differentiable due to the discrete ranking that involves an indicator function. To make it smooth and enable meaningful gradient computation for feature learning, the fastAP loss removes sorting entirely and introduces a distance-quantization technique. In other words, the precision and recall in AP as in,
>
> $$
> \mathrm{AP} = \int_{\Omega} \mathrm{Prec}(z)\, d\mathrm{Rec}(z),
> $$
>
> are approximated with smooth functions of the distance between query and database embeddings. $z\in\Omega$ denotes the continuous distance values between query and database feature representations. At a high level, these distances are uniformly divided into bins, and the database images are assigned to the corresponding bins based on distance scores. Upon solving the above equation using distance distributions, we obtain the final fastAP expression in Eq. (9) of the main paper, which approximates the AP. In the final expression,  $H_j^+$ is the cumulative positive retrievals until the bin $j$, $h_j^+$ denotes the number of positive retrievals in the $j^{th}$ bin, $H_j$ computes the total cumulative retrievals until the $j^{th}$ bin, and $N_q^+$ denotes the total number of positives for the query $q$. In addition, our approach incorporates the flipped roles of the query and positives, resulting in Eq. (10) in the main paper.
>
> **Requested Changes 3**: Notations $P^{\prime}_q$
>
> **Response**: They both still convey the same meaning. The difference is in the positive sets. Eq (2) is a more general expression, where $P^{\prime}_q=P_q\cup\{q\}$ implies a comprehensive set that contains the query and all of its positives. Similarly, Eq (10), $P_q^\prime=\{p,q\}$ (which is essentially $\{p\}\cup\{q\}$), also defines a set consisting of a query and its positives. The triplet loss contains a single positive and a negative for a query. Therefore, Eq (2), when applied to a triplet, gives us Eq (10). Nevertheless, for consistency, we follow the same notation in both equations.
>
> **Requested Changes 4**:  We thank you for pointing grammatical errors. We have made the suggested changes in the manuscript.
>
> **Requested Changes 5**: Confusion between absolute value notation and a hinge function
>
> **Response**: Thank you for this comment. There is no $q_{qp}$ notation in the loss function in Eq. (9). $d_{qp}$ and $d_{qn}$ imply the distance metric between query-positive (q, p) and query and negative (q, n) embeddings. Further, \$\|.\|\_\+\$ is followed from the MS-NetVLAD paper. We will change it to \$\[.\]\_\+\$, where this defines max(., 0). We hope this now avoids the confusion between absolute value and hinge constraint.

---

### Review · Reviewer_u1rk · 2026-06-12

**Summary Of Contributions:**

This paper proposes ANU-RL, a simple modification to metric-learning objectives for visual place recognition. Instead of optimizing only query-positive and query-negative relations, the method expands each query neighborhood by treating the query and its positives as a set of anchors. Each anchor is then pulled toward the remaining positives and pushed away from negatives. The authors instantiate this idea for Multi-Similarity, Triplet, and FastAP losses, and evaluate it with several VPR aggregators including BoQ, SALAD, MixVPR, NetVLAD, CosPlace, ConvAP, R2Former-GR, and SuperVLAD. The paper also studies easy/hard/all variants of the added relations and includes an additional image-retrieval experiment.

The main strength is that the idea is intuitive, easy to implement, and does not add inference-time cost. The experimental section is broad for VPR: it covers many aggregators, several challenging datasets, multiple loss functions, and some qualitative analysis. The reported gains on difficult settings such as Tokyo 24/7, Nordland, and SVOX Night suggest that explicitly using positive-positive and positive-negative relations can improve representation quality.

The main weakness is that the claims are stronger than what the evidence fully supports. The paper repeatedly frames the method as unsupervised and broadly plug-and-play for contrastive learning, but the training still relies on positive/negative neighborhood definitions derived from place labels or geo-supervision, and the generality claim is only partially tested. The method is also closely related to existing multi-positive metric learning and supervised contrastive ideas, but the novelty over these directions is not sharply separated. In addition, some equations and notation appear inconsistent, and the loss normalization/scale is not sufficiently controlled, making it hard to disentangle the proposed relational structure from simply adding more pair terms and stronger gradients.

**Additional Comments:**

The idea is simple and practically useful, and the experiments suggest that adding positive-positive and positive-negative relations can improve VPR representations. However, the current version needs clearer claims, corrected formulas, stronger controls, and a more careful discussion of supervision and variant selection. I would be more positive if the authors revise the framing from broad “unsupervised/SOTA/any contrastive loss” claims to a more precise and well-supported contribution: a practical multi-positive anchor augmentation framework for VPR metric learning with no inference-time overhead.
A few minor presentation and related-work suggestions are listed below.

1.	The figures appear to be raster images rather than vector graphics. Please convert them to vector graphics to improve readability, especially for mathematical diagrams, t-SNE plots, and tables.
2.	If the authors intend to broaden the related-work discussion toward recent visual foundation models or diffusion-based visual priors, they may consider adding relevant research papers on the foundations and applications of diffusion models, such as [A], [B], and [C]. This suggestion is optional, since the main focus of the paper is VPR metric learning rather than diffusion modeling.

References:

[A] Tony Bonnaire, Raphaël Urfin, Giulio Biroli, and Marc Mézard. Why Diffusion Models Don’t Memorize: The Role of Implicit Dynamical Regularization in Training. NeurIPS 2025.

[B] Shuhai Zhang, Feng Liu, Jiahao Yang, Yifan Yang, Changsheng Li, Bo Han, and Mingkui Tan. Detecting Adversarial Data by Probing Multiple Perturbations Using Expected Perturbation Score. ICML 2023.

[C] Jing He, Haodong Li, Wei Yin, Yixun Liang, Leheng Li, Kaiqiang Zhou, Hongbo Zhang, Bingbing Liu, and Ying-Cong Chen. Lotus: Diffusion-based Visual Foundation Model for High-quality Dense Prediction. ICLR 2025.

**Audience:**

Yes

**Audience Explanation:**

The findings should be interesting to part of the TMLR audience, especially readers working on visual place recognition, image retrieval, metric learning, and representation learning. The paper addresses a practical question: when multiple positives are naturally available for a query, should training objectives explicitly model relations among those positives rather than only between the original query and its positives/negatives? The answer appears to be often yes, particularly for challenging VPR datasets with viewpoint, illumination, seasonal, and temporal shifts.

The method is also attractive because it is simple and has no inference-time overhead. Even if the novelty is incremental relative to existing multi-positive metric-learning ideas, the broad VPR evaluation and the integration with several modern aggregators make the empirical study useful. The work would be more valuable if the authors clarified the supervision setting, tightened the claims, fixed the formalism, and provided stronger controls for loss scaling and variant selection.

**Broader Impact Concerns:**

The paper includes a broader impact discussion, but it should be expanded. Improved visual geo-localization can benefit robotics, navigation, and localization in GPS-denied environments. However, VPR and image retrieval can also be used for surveillance, tracking, and privacy-sensitive localization. The paper should explicitly discuss potential misuse in surveillance and large-scale location inference.
The paper should also discuss safety risks in autonomous navigation. The authors acknowledge that some aggregators show inconsistent performance and that failure modes are not fully theoretically understood. Since VPR can be used in risk-sensitive systems, this limitation should be connected more directly to deployment safety, uncertainty estimation, and the need for fallback mechanisms.

**Claims And Evidence:**

No

**Claims Explanation:**

The empirical evidence supports a useful and promising training modification, but it does not fully support the strongest claims in the paper.

1.	The claim of achieving state-of-the-art performance is not sufficiently established. The main tables compare models trained with the proposed loss against reproduced baselines under the authors' own training setup. The paper itself notes that some reproduced baselines are below the published off-the-shelf results due to different input resolution, batch size, and hyperparameters. Therefore, the results convincingly show improvement over the authors' reproduced baselines, but not necessarily SOTA under standard or best-known settings.

2.	 The “unsupervised representation learning” framing is unclear. The proposed loss requires positives and negatives for each query. In VPR, these are typically obtained from geo-localization labels or place-level supervision. The paper should clarify whether the setting is supervised, weakly supervised, self-supervised, or unsupervised, and should align the title and claims accordingly.

3.	The evidence for broad plug-and-play generality is only partial. The method is tested mainly on VPR with MSim, and appendix experiments cover Triplet and FastAP. The image-retrieval extension is encouraging, but the paper excludes MSim image-retrieval results because the changes are minor. Thus, the claim that the framework applies generally to any contrastive loss should be softened or supported by more systematic experiments.
4.	Several methodological details need clarification before the evidence can be considered fully convincing. The added positive-anchor loop changes the number of pair terms and the loss scale, but the paper does not provide a clear normalization control showing that the gains are not simply due to stronger gradients or an effective learning-rate change. The selection among ANU-ALL, ANU-Hardest, and ANU-Easiest is also not fully justified by a validation protocol, which raises the concern that the best variant may be chosen using test-set performance.
5.	The paper contains notation issues in the formulas for the hard/easy variants and the generalized losses. These do not necessarily invalidate the implementation, but they reduce confidence in the formal presentation and should be corrected.

**Requested Changes:**

Critical:

1.	Clarify the learning setting. The paper should explain why the method is called unsupervised despite relying on positive and negative neighborhoods. If the positives are derived from geo-tags or place labels, the title and claims should be revised to “weakly supervised” or “metric learning for VPR” unless a truly unsupervised construction is used.
2.	Temper or substantiate the SOTA claim. The paper should either compare against published SOTA models under standard settings or rephrase the claim as improvement over reproduced baselines under the authors' training protocol. This is important because the reproduced baselines are not always the published best results.
3.	Fix the mathematical notation. In the hard/easy variants, the negative index appears to be selected from the positive set rather than from the negative set. The Triplet and FastAP extensions also need clearer definitions of anchors, positives, negatives, histogram terms, and summation indices. These equations should be checked against the actual implementation.
4.	Add a normalization and loss-scale control. Since ANU-RL introduces many additional pair terms while retaining a similar outer normalization, it may change the gradient scale. The authors should compare against a normalized version, a reweighted baseline, or a matched-gradient/matched-learning-rate control to show that the gain comes from the proposed relations rather than from a larger effective optimization signal.
5.	Define a clear variant-selection protocol. The paper should state whether ANU-ALL, ANU-Hardest, or ANU-Easiest is selected per model, per dataset, or fixed before evaluation. A validation-based rule is needed to avoid choosing variants based on test-set performance.
6.	Report statistical reliability. The main VPR results should include multiple seeds or confidence intervals for at least representative models and datasets. Without this, it is hard to judge whether small improvements or drops are meaningful.

Strengthen:

1.	Strengthen the comparison to related metric-learning objectives such as supervised contrastive learning, N-pair loss, lifted structured loss, batch-hard triplet variants, and proxy-based methods. The paper should explain more precisely what is new beyond using all positives as anchors.
2.	Give a clearer complexity analysis in the main paper. The appendix shows increased training time for MSim-based variants, but the main text emphasizes zero inference cost. The training-time trade-off should be stated more prominently.
3.	Provide aggregated results across datasets and models. The current tables are large, but it is difficult to see the average behavior, failure cases, and sensitivity to model capacity. A summary table with average R@1/R@5 gains and number of wins/losses would help.
4.	Analyze failure modes more concretely. The method assumes that all positives should be mutually close, but VPR positives may be multi-modal due to viewpoint, lighting, occlusion, or temporal changes. The paper should discuss when pulling positives together may hurt.
5.	Improve writing and proofreading. There are several typos and awkward phrases, such as “can, form,” “Easiset,” “Pacth-NetVLAD,” “Triple loss,” and inconsistent use of “its/it's.” Cleaning these issues would make the paper more professional.

---

> ### Author Response · Authors · 2026-07-06
> **In continuation to the above discussionThis response addresses some of the concerns raised by the reviewer, such as scaling the additional terms, averaging recall performance to ensure statistical stability, and the ANU variant selection protocol.**
>
> We thank the reviewer for the careful evaluation of our work and insightful comments. Addressing these comments significantly improved the quality of the work.  We provide a detailed response to the comments as follows.
>
> **Response to the summary of the weaknesses identified**: Responses to comments on **Unsupervised setting** and **Plug-and-play** are discussed in the subsequent comment boxes.
>
> About **generalizability**, we agree with the reviewer that we need to test the proposed framework for various other contrastive loss functions. However, most existing ones follow a common approach as expressed by Eq. 1 in the main paper. We expect the proposed approach to be particularly helpful in complex scenarios where images convey the same semantics but appear different. Nevertheless, as mentioned in the future work section (Section 6) of the main paper, we will extend our research to various loss functions, as the reviewer noted.
>
> Regarding **novelty**, the evolution of contrastive loss functions has been built on incremental changes rather than on something entirely new. Some widely used loss functions are broadly related as follows.
>
> $$
> L_{\text{contrast}}
> \subseteq
> L_{\text{triplet}}
> \subseteq
> L_{\text{in-trip-mine}}
> \subseteq
> L_{\text{lsl}}
> \subseteq
> L_{\text{n-pair}}
> \subseteq
> L_{\text{GLS-BA}}
> \subseteq
> L_{\text{sup-con}}
> \subseteq
> L_{\text{ms}}
> \subseteq
> L_{\text{ANU-RL}}.
> $$
>
> The early contrastive loss, $L_{\text{contrast}}$, operates with pairs, positives, and negatives in different updates independently. The subsequent work, $L_{\text{triplet}}$ loss, combines both positives and negative pairs, and optimizes the loss with both in a single update; hence, it could handle the problem that $L_{\text{contrast}}$ tries to address. Further, In-triplet mining, $L_{in-trip-mine}$, introduces an additional distance $d_{pn}$ connecting positive to negative. The final expression includes either $d_{an}$ or $d_{pn}$, whichever is harder. Effectively dealing with triplets, which could match with the triplet loss if the $d_{pn}$ is less informative. For example, during the cross-seasons scenario, say the Nordland dataset, positive and negative images come from the same season, summer, while the query is from winter. In that case, $d_{an}$ is harder than $d_{pn}$, reducing the loss to the triplet loss. The Lifted-structure loss, $L_{lsl-hard}$, the harder version of it, is closely related to the $L_{in-trip-mine}$ loss. The difference is that the $L_{lsl-hard}$ samples the hardest pairs within the batch. The smoother version of it, $L_{lsl-smooth}$, includes all pair-wise connections, including positive to negatives. This inherently solves the problem that $L_{in-trip-mine}$ tries to address. In contrast, the N-Pairs loss, $L_{n-pair}$, extends the triplet variants to incorporate multiple negatives. The Generalized Lifted Structure loss (GLS) with the Batch-All case, $L_{GLS-BA}$, extends the triplet loss, which proposes an efficient pair mining strategy, exploiting same-class pairs as positives and samples from all other classes as negatives. The subsequent work, Supervised Contrastive Loss, $L_{sup-con}$, does the same job but with different expressions. Similarly, a contemporary work with SupCon loss, the Multi-Similarity loss, $L_{ms}$, closely follows the $L_{GLS-BA}$ loss with a minor difference. Broadly, we can classify these loss functions based on the relationships they use as follows. 1. Single positive and negative pairs, 2. single positive and multiple negatives, 3. multiple positives and multiple negatives, and 4. the proposed approach introduces a new class with all possible inter-connections between them. If we observe closely, these works evolved with minor improvements over the preceding ones.
>
> Thank you for spotting the notational inconsistency. We improved the writing quality by incorporating the suggested changes in the revision.
>
> Response to the comment on **normalization**/**scale matching** are provided in the following comment boxes as this also appears in the suggested critical changes.

---

> ### Author Response · Authors · 2026-07-06
> **In continuation to the above discussion**
>
> **Weakness 1**:  Regarding the SOTA claim, we agree with the reviewer that the reproduced results do not match the SOTA results. Due to limited resources, we cannot use the settings of the baseline approaches. Moreover, we used SOTA to refer to the recent models, not to state that we achieved SOTA performance. If you could point us to the lines that read like achieving SOTA performance, we would change them as suggested.
>
> **Weakness 2**: Many thanks for the comment on Unsupervised. We rephrased it to **weakly supervised**. It makes sense. As the reviewer noted, we rely on positives and negatives which offer weak form of supervision for the model training, which does not fall under the unsupervised scenario.
>
> **Weakness 3**: **Plug-and-play** claim is almost always viewed in terms of compatibility. Most approaches that make such a claim do not guarantee performance gains. As you noted, whether it helps or hurts requires an empirical investigation, which is exhaustive, since the family of contrastive losses is large and many applications rely on them. The other way is to come up with mathematical guarantees, which is also not trivial without assumptions. Nevertheless, it will be interesting to incorporate various other loss functions into the proposed framework and experiment with them; we consider this for future work due to the short rebuttal window. MSim image-retrieval results are in Appendix A.4.1 (revised) and presented in **Table 2**. Nevertheless, we made it clear in the revised manuscript that our future research will broaden generalizability tests by extending the proposed approach to other contrastive loss functions suggested by the reviewer.
>
> **Table 2.**  All experiments are average over 5 runs. The proposed framework is particularly helpful in cases of domain shift such as Tokyo 24/7, Nordland, and SVOX Night datasets with severe appearance variation. In other cases, the gains are small, as we see in this table.
>
> | Dim | CUB-200-2011 (MSim) | CUB-200-2011 (ANU-MSim) | CARS-196 (MSim) | CARS-196 (ANU-MSim) | SOP (MSim) | SOP (ANU-MSim) |
> | ---: | ------------------: | ----------------------: | --------------: | ------------------: | ---------: | -------------: |
> | 64  | 57.26 | **57.30** | 66.75 | **65.85** | 59.78 | **60.03** |
> | 128 | 61.49 | **61.66** | **73.91** | 72.07 | 65.15 | **65.32** |
> | 512 | 66.09 | **66.45** | **80.79** | 80.21 | 71.39 | **72.53** |
>
> **Requested Changes-Critical 1**: Please refer to  **Weakness 2**
>
> **Requested Changes-Critical 2**: Please refer to **Weakness 1**
>
> **Requested Changes-Critical 3**: Explanation for some of the under-defined terms in the main paper are included in Appendix A.4.1. and corrections are made
>
> **Requested Changes-Critical 4**: Appendix A.7 has been revised to discuss the impact of normalization. Most losses with the same hyperparameters often vary in scale. In our work, the Multi-Sim, Triplet, and FastAP losses show different loss scales. Therefore, we do not see it as necessary but rather as a means of maintaining numerical stability, which the proposed losses do not run into. Regardless, we present limited results for normalized ALL and AH in **Table 3**. Extended results are in the revised manuscript. We see a drop in most cases compared to un-normalized cases. This could be because the normalization resets the strength of the additional terms for optimization. When we view ANU-RL as a regularization technique, normalizing is similar to choosing one of the possible values of $\lambda$. The best $\lambda$ in standard techniques like L1/L2 regularizers is often selected empirically. In our experiments, we used $\lambda=1$. Nevertheless, future research can sweep over a range of $\lambda$ values and select the best one based on the downstream application. Further, we observe no significant scale difference between baseline and our losses. The baseline training begins with a loss magnitude of 1, and ours begins with 2.5, for the same set of hyperparameters, which is well within the acceptable margin. These training plots are presented in Fig. 12 of the revised version.

---

> ### Author Response · Authors · 2026-07-06
> **This is in continuation to the above discussion**
>
> **Table 3.** Performance analysis with normalization of the proposed ANU-RL variants. We notice a drop in performance in most cases with the proposed variants compared to unnormalized cases. Scaling the loss function is similar to scaling the regularization term. The drop in performance in this case could be because of the wrong choice of the $\lambda$ in our framework.
>
> | Method | ANU Pairs | Tokyo 24/7 | Nordland | MSLS (Val) | Amstertime | SPEDTest | Eynsham | SVOX Night |
> |--------|-----------|-----------:|----------:|-----------:|-----------:|----------:|---------:|-----------:|
> | BoQ | ALL-Norm | 80.95 | 67.66 | 84.86 | 35.85 | 78.75 | 87.86 | 58.57 |
> | BoQ | AH-Norm | **81.59** | **73.54** | **85.54** | **38.94** | **82.04** | **89.29** | 60.27 |
> | MixVPR | ALL-Norm | 79.05 | 72.41 | 82.97 | 35.37 | 83.53 | 87.79 | **65.25** |
> | MixVPR | AH-Norm | **79.68** | 72.72 | 81.89 | **36.18** | **84.68** | **87.94** | 57.47 |
> | CosPlace | ALL-Norm | 68.89 | 58.59 | 80.27 | **30.73** | 77.59 | 87.09 | **65.01** |
> | CosPlace | AH-Norm | **70.79** | 58.34 | 78.11 | 29.67 | **77.92** | **87.28** | 44.59 |
> | ConvAP | ALL-Norm | **77.46** | **63.92** | **76.62** | **34.39** | 81.55 | 86.04 | 52.00 |
> | ConvAP | AH-Norm | 75.56 | 62.61 | 74.32 | 33.41 | **83.53** | **86.10** | 50.91 |
>
> **Requested Changes-Critical 5**: In the majority of the cases, ANU-ALL and ANU-Hardest both outperform the baseline approaches. AH (AE) involves only the hardest (the easiest) similarity term, while ALL involves all multiple similarities. Therefore, AH/AE involves slightly reduced computation; hence, we recommend AH, the best-performing variant among AH and AE.
>
> However, among AH and ALL, AH works well in some cases and ALL in other cases. To balance fluctuations and leverage the combined capabilities of both variants, we introduce additional parameters in the framework and try it in our future work, This is included in Appendix A.8 of the revised manuscript.
>
> **Requested Changes-Critical 6**: Representative results are presented in **Table 4**, where we observe the consistent trend as in the main paper.
>
> **Table 4.** R@1 performance averaged over three runs with different seeds. The performance trends are consistent with those reported in the main paper.
>
> | Method | Variant | ANU pairs | Tokyo 24/7 | Nordland | MSLS (Val) | SVOX Night |
> |--------|---------|----------:|-----------:|----------:|-----------:|-----------:|
> | **BoQ** | BL-Avg | 81.06 ± 1.28 | 72.44 ± 1.81 | 85.72 ± 0.69 | 37.62 ± 0.68 | 61.04 ± 4.72 |
> |  | ALL-Avg | 80.32 ± 2.51 | 71.77 ± 0.65 | 85.54 ± 0.81 | 36.89 ± 1.37 | 61.93 ± 5.27 |
> |  | **AH-Avg** | **84.55 ± 1.43** | **77.18 ± 1.56** | **86.49 ± 0.48** | **39.84 ± 2.12** | **67.00 ± 3.93** |
> | **MixVPR** | BL-Avg | 77.99 ± 1.20 | 72.91 ± 1.96 | 83.02 ± 0.55 | 35.42 ± 0.76 | 46.54 ± 2.49 |
> |  | ALL-Avg | 77.77 ± 1.68 | 74.36 ± 1.67 | **84.27 ± 0.21** | **37.39 ± 0.29** | 49.98 ± 2.00 |
> |  | **AH-Avg** | **80.63 ± 0.32** | **77.23 ± 1.14** | 84.19 ± 1.02 | 37.32 ± 0.08 | **50.59 ± 0.91** |
> | **CosPlace** | BL-Avg | 68.67 ± 1.47 | 59.66 ± 0.82 | 80.36 ± 0.52 | 29.56 ± 0.57 | 36.90 ± 4.10 |
> |  | **ALL-Avg** | 71.01 ± 1.85 | **63.66 ± 1.18** | **81.80 ± 0.39** | **30.79 ± 1.26** | **42.21 ± 2.90** |
> |  | AH-Avg | **71.54 ± 1.31** | 63.12 ± 1.54 | 81.49 ± 0.36 | 29.84 ± 0.57 | 35.52 ± 1.21 |
> | **ConvAP** | BL-Avg | 76.30 ± 2.42 | 64.38 ± 1.48 | 78.15 ± 1.67 | 33.22 ± 0.20 | 24.18 ± 4.36 |
> |  | ALL-Avg | 75.56 ± 1.27 | 65.47 ± 0.37 | **80.09 ± 0.41** | **34.77 ± 0.73** | **30.01 ± 3.27** |
> |  | **AH-Avg** | **78.62 ± 0.96** | **66.90 ± 0.70** | 77.48 ± 1.01 | 33.52 ± 0.13 | 19.56 ± 0.84 |
>
> **Requested Changes-Strengthen 1**: Due to time constraints for the rebuttal, we are unable to run our approach on the suggested losses. We will consider running these experiments in our future research as mentioned in Section 6.
>
> **Requested Changes-Strengthen 2**: Yes, we agree with the reviewer that the training time increases for our approach. During training, additional relationships require additional computations, increasing training time. We used loop-based scripts in our implementation. However, advanced dynamic programming concepts can parallelize and speed up the computations. As claimed, once training finishes, the objective function plays no role in inference. Hence, no change in the inference time.
>
> **Requested Changes-Strengthen 3**: Please refer to Table 13 in Appendix of the revised draft and a subset of results in **Table 5** in the following comment box.

---

> ### Author Response · Authors · 2026-07-06
> **This is in continuation to the above discussion**
>
> **Table 5.** Average R@1 and R@5 performance across datasets. The proposed AH and ALL variants achieve the best performance in the majority of cases.
>
> | Model | ANU-Type | Avg R@1 | Avg R@5 | # Wins Ratio (R@1) | # Wins Ratio (R@5) |
> |------|----------|--------:|--------:|-------------------:|-------------------:|
> | **BoQ** | BL | 80.72 | 89.56 | 2/14 | 5/14 |
> |  | ALL | 79.70 | 88.59 | 0/14 | 0/14 |
> |  | **AH** | **82.64** | **90.81** | **13/14** | **9/14** |
> |  | AE | 76.17 | 86.63 | 0/14 | 0/14 |
> | **SALAD** | BL | 87.63 | 94.80 | 4/14 | 2/14 |
> |  | ALL | 87.47 | 94.54 | 1/14 | 1/14 |
> |  | **AH** | **88.17** | **95.07** | **8/14** | **13/14** |
> |  | AE | 85.24 | 92.94 | 1/14 | 0/14 |
> | **MixVPR** | BL | 78.51 | 87.98 | 1/14 | 1/14 |
> |  | ALL | 78.89 | 88.78 | 5/14 | 6/14 |
> |  | **AH** | **79.73** | **88.84** | **7/14** | **8/14** |
> |  | AE | 74.79 | 85.99 | 1/14 | 0/14 |
> | **ConvAP** | BL | 69.24 | 80.97 | 0/14 | 1/14 |
> |  | **ALL** | **72.83** | **82.82** | **10/14** | **11/14** |
> |  | AH | 70.66 | 82.73 | 4/14 | 1/14 |
> |  | AE | 68.36 | 81.97 | 0/14 | 1/14 |
>
>
> **Requested Changes-Strengthen 4**:  From the formulation, our framework seems to help in such complex scenarios due to additional relationships. This could be because, we show the model all possible relationships between these positive images, while contrasting them from the confusing negative images. However, the training dataset do not contain all the mentioned challenging scenarios in equal distribution, hence, the models still suffer from domain shifts. For example, we see that most models achieving above 90\% R@1 on Pittsburgh like datasets often fail to perform competitively on the Tokyo, Nordland, SVOX-Night like challenging datasets. Moreover, this is not specific to our approach. Baseline approaches also suffer from this.
>
> **Requested Changes-Strengthen 5** Many thanks to the reviewer for identifying these typos. We have corrected all of these in the revised version of the manuscript.
>
> **Suggested broader impact concerns are incorporated in Appendix A.1. of the revised paper.**
>
> **Additional Comments**  Thank you for the suggestions. We will make these changes in the final version of the manuscript.

---

### Review · Reviewer_VfYD · 2026-06-23

**Summary Of Contributions:**

This paper proposes ANU-RL, a plug-and-play framework that enhances contrastive representation learning by explicitly modeling additional intra-positive and positive–negative relationships. The method is applied to multiple standard losses (e.g., MSim, Triplet, FastAP) and integrated into several state-of-the-art VPR aggregators, consistently improving performance across a range of benchmarks without introducing inference-time overhead. Key strengths include its general applicability, simplicity, and consistent empirical gains across different architectures and datasets. A main limitation is that the theoretical justification and broader analysis of its robustness and applicability remain somewhat limited.

**Audience:**

Yes

**Audience Explanation:**

This paper would be of interest to at least some members of the TMLR audience, particularly researchers working on metric learning, contrastive representation learning, and visual place recognition (VPR). The proposed ANU-RL framework offers a simple and general modification to existing contrastive losses by incorporating additional intra-positive and positive–negative relationships, and demonstrates consistent empirical improvements across multiple VPR architectures and benchmarks. Given its plug-and-play nature, broad applicability to common losses (MSim, Triplet, FastAP), and extension to image retrieval tasks, the findings are relevant to both methodological researchers and practitioners seeking improved representation learning techniques.

**Broader Impact Concerns:**

The Broader Impact section is well written and complete, clearly distinguishing between positive and negative impacts. It appropriately discusses potential risks in safety-critical applications such as autonomous driving, robotic navigation, and surveillance, and also acknowledges the limitations in terms of unstable performance and incomplete theoretical understanding. Overall, it meets the standard requirements for a Broader Impact statement in TMLR, and no significant issues are identified.

**Claims And Evidence:**

Yes

**Claims Explanation:**

The claims are generally supported by accurate and convincing evidence. The paper provides extensive empirical validation across multiple VPR architectures (e.g., BoQ, SALAD, MixVPR, NetVLAD) and diverse benchmark datasets, consistently demonstrating performance improvements over strong baselines. In addition, the authors include comprehensive ablation studies (ANU-ALL, ANU-Hard, ANU-Easy), qualitative visualizations, complexity analysis, and theoretical interpretation in the appendix, all of which strengthen the credibility of the proposed framework. The improvements are also shown to generalize to image retrieval tasks without introducing inference-time overhead. However, while the empirical evidence is strong and consistent, the mechanistic explanation for the performance gains remains partially qualitative, and the contribution of different interaction types is not fully disentangled.

**Requested Changes:**

1. Appendix A.3 reports results on an NVIDIA RTX 3090. For reproducibility, it would be helpful to clarify whether all experiments were conducted under the same hardware setup, and briefly discuss how computational cost scales with the number of positive samples per query.

2. While 224×224 inputs ensure fair comparison, evaluating or discussing higher-resolution settings would further strengthen the analysis and verify whether performance gains remain consistent.

3. Table presentation could be improved by standardizing numerical precision (e.g., consistent decimal places) for better readability.

4. A brief discussion on the scenarios where ANU-RL is most effective would further improve the practical insights of the paper.

---

> ### Author Response · Authors · 2026-07-05
> **Our response clarifies the hardware used for experiments, analysis on multiple input resolutions, and the effectiveness of the proposed ANU-RL work**
>
> We thank the reviewer for the positive assessment of our work and thoughtful comments. Our responses to the comments are as follows.
>
> **Comment**: Limited theoretical justification and broader analysis of its robustness and applicability.
>
> **Response**: We agree that there is no theoretical guarantee we could provide at this point. However, one way we could interpret the proposed loss function as a **regularizer** is as shown in section 3.4 of the main paper. The impact of the additional relationships that mimic the regularization term is shown through extensive experiments on diverse datasets, which, in most cases, improve over the baseline objectives. Nevertheless, we will continue to unfold the proposed approach and understand it theoretically in our future research.
>
> Regarding **robustness and applicability**, from empirical analysis, we notice that the proposed variants, AH and ALL seem to improve over challenging scenarios. For instance, the BoQ+AH, improves by around 3 points on the Nordland, 5 points on the Tokyo 24/7, 2 points on the Amstertime, and 6 points on the SVOX-Night datasets. This is from the average recalls over three runs, included in Appendix A.9 in the revised manuscript.
>
> **Comment**: The mechanistic explanation for the performance gains remains partially qualitative, and the contribution of different interaction types is not fully disentangled.
>
> **Response**: Intuitively, the proposed approach is helping; this could be because existing approaches are mono-directional, looking from the query point of view alone. In contrast, our work makes it bi-directional. In cases where there are viewpoint, seasonal, illumination, etc., changes between a query and its positives, these bidirectional relationships enable the model to capture unified features that are robust to these variations by explicitly matching images under these challenging conditions. However, we agree that a better interpretation is needed rather than intuition. We follow the suggestion and continue our research to establish the mathematical significance of the proposed relationships.
>
> **Requested Changes 1**: The complexity analysis is performed entirely on the NVIDIA RTX 3090 GPU. However, to speed up the experiments, we split them across different machines, including the RTX 3090 and L40S.
>
> In our implementation, we follow a sequential approach by looping over the positives for each query; hence, the memory complexity remains the same. Time complexity scales as O(N), where N is the number of positives for a query ($P_q$) or the number of images sampled from each city. Compared to the BL, we repeat the loss computation for $P_q$ times; hence, it is O(N).
>
> **Requested Changes 2**: We thank you for this suggestion. Some of the aggregators used in this work cannot be extended directly to images of arbitrary dimensions because MLPs (e.g., MixVPR) operate along the spatial dimensions, leading to compatibility issues. Instead, we evaluate the performance of the a few representative algorithms across different input dimensions. These results are shown in **Table 1** below. We present here a subset of results in Tables 15 and 16 in the revised manuscript.
>
> **Table 1**: Performance comparison of VPR models on varying input dimensions. We observe that as the image resolution increases, the proposed approach continues to improve.
> | Method | ANU pairs | Tokyo 24/7 | Nordland| MSLS (Val) ) | Amstertime | SVOX Night |
> |--------|-----------|-----------------:|---------------:|-----------------:|-----------------:|-----------------:|
> | BoQ | BL-320 | 85.71 | 75.01 | 87.70 | 40.16 | 68.89 |
> | BoQ | ALL-320 | 82.54 | 73.16 | 87.03 | 36.75 | 69.14 |
> | BoQ | AH-320 | **87.30** | **80.62** | **88.11** | **42.11** | **71.20** |
> | --- | --- | --- | --- | --- | --- | --- |
> | SALAD | BL-448 | 94.60 | 88.09 | 92.16 | 59.35 | 96.84 |
> | SALAD | ALL-448 | 95.56 | **88.89** | 91.89 | 59.43 | 96.72 |
> | SALAD | AH-448 | **96.51** | 88.83 | **93.24** | **63.01** | **97.33** |
> | --- | --- | --- | --- | --- | --- | --- |
> | CosPlace | BL-320 | 72.70 | 61.99 | 82.97 | 32.85 | 35.84 |
> | CosPlace | ALL-320 | 75.24 | **66.41** | **84.73** | **33.09** | **49.45** |
> | CosPlace | AH-320 | **76.83** | 62.69 | 83.92 | 32.03 | 44.47 |
> | --- | --- | --- | --- | --- | --- | --- |
> | ConvAP | BL-320 | 77.78 | 65.16 | 76.62 | 36.18 | 14.95 |
> | ConvAP | ALL-320 | 79.05 | **68.28** | **80.14** | **36.26** | **23.45** |
> | ConvAP | AH-320 | **79.68** | 68.16 | 78.24 | 34.96 | 17.13 |
>
> **Requested Changes 3**: Many thanks for the suggestion. We adjusted decimal places in the revised submission.
>
> **Requested Changes 4**: From the empirical performance, it appears to help with appearance variation caused by day-night shifts and seasonal changes. The corresponding datasets include Nordland, Tokyo, Amstertime, and the SVOX-Night, where we see that the losses under the proposed framework show significant gains in the majority of cases.